# Iron deficiency causes aspartate-sensitive dysfunction in CD8+ T cells

Megan R. Teh [1,8] ✉, Nancy Gudgeon [2], Joe N. Frost [1], Linda V. Sinclair [3], Alastair L. Smith [4], Christopher L. Millington [4], Barbara Kronsteiner [5], Jennie Roberts[6], Bryan P. Marzullo [6], Hannah Murray[1], Alexandra E. Preston [1], Victoria Stavrou [2], Jan Rehwinkel[1], Thomas A. Milne [4], Daniel A. Tennant [6], Susanna J. Dunachie [5,7], Andrew E. Armitage [1], Sarah Dimeloe [2,9] & Hal Drakesmith [1,9] ✉

Iron is an irreplaceable co-factor for metabolism. Iron deficiency affects >1 billion people and decreased iron availability impairs immunity. Nevertheless, how iron deprivation impacts immune cell function remains poorly characterised. We interrogate how physiologically low iron availability affects CD8+ T cell metabolism and function, using multi-omic and metabolic labelling approaches. Iron limitation does not substantially alter initial post-activation increases in cell size and CD25 upregulation. However, low iron profoundly stalls proliferation (without influencing cell viability), alters histone methylation status, gene expression, and disrupts mitochondrial membrane potential. Glucose and glutamine metabolism in the TCA cycle is limited and partially reverses to a reductive trajectory. Previous studies identified mitochondria-derived aspartate as crucial for proliferation of transformed cells. Despite aberrant TCA cycling, aspartate is increased in stalled iron deficient CD8+ T cells but is not utilised for nucleotide synthesis, likely due to trapping within depolarised mitochondria. Exogenous aspartate markedly rescues expansion and some functions of severely iron-deficient CD8+ T cells. Overall, iron scarcity creates a mitochondrial-located metabolic bottleneck, which is bypassed by supplying inhibited biochemical processes with aspartate. These findings reveal molecular consequences of iron deficiency for CD8+ T cell function, providing mechanistic insight into the basis for immune impairment during iron deficiency.

Studies of metabolism in cancer and in different cell lineages have changed understanding of how metabolic processes regulate cellular transformation and cell fate[1,2]. In contrast, how common nutritional deficiencies influence metabolic pathways and alter cell biology is relatively poorly studied. Iron deficiency is the most common micronutrient deficiency worldwide[3], inhibiting erythropoiesis, impairing cognitive development and disabling immunity, and is a comorbidity for several disorders, including heart failure[4–6].

Within cells, iron is utilised by ~2% of proteins and ~6.5% of enzymes (including ~35% of oxidoreductases)[7]. Key conserved biochemical processes are iron-dependent, such as the electron transport chain (ETC), tricarboxylic acid (TCA) cycle, nucleotide synthesis, DNA repair, histone and DNA demethylation and oxygen sensing[6,7]. Cells typically obtain iron by receptor-mediated internalisation of transferrin, a dedicated iron-chaperone protein in plasma and extracellular fluid[8]. However, the amount of iron bound to transferrin shows

marked physiological variation. For example, in severe iron deficiency or in the presence of inflammation, plasma iron concentrations can drop by ~90%[9,10], drastically decreasing iron supply to cells.

Notably, human genetic studies and pre-clinical work link decreased transferrin-iron acquisition by proliferating lymphocytes to impaired adaptive immunity[5,11]. Patients homozygous for a mutation in *Tfrc* (the gene encoding the transferrin receptor, TFR1, also known as CD71) that inhibits cellular iron uptake, have a severe combined immunodeficiency characterised by reduced lymphocyte function and suppressed antibody titres[11]. Similarly, low serum iron availability suppresses B and T cell responses in mice following vaccination and influenza infection[5]. How iron deficiency mechanistically impairs adaptive immune cells remains unclear, but understanding this issue could inform how iron regulates immunity and provide further rationale for correcting iron deficiency within human populations. Recent information summarising how iron influenced the immune system has been reviewed here[12].

T cells dramatically remodel their metabolism post-activation, including upregulation of glycolysis and oxidative phosphorylation (OXPHOS), in order to meet the increased energetic and biosynthetic requirements of proliferation and effector function[13]. Transition from naïve to activated T cell is accompanied by greatly increased expression of the iron uptake protein, TFR1 and CD8[+] T cell iron content is calculated to triple within the first 24 h post-activation[14]. In this study, we investigate how low iron supply influences activated CD8[+] T cells via 'omics approaches and isotope tracing. We find mitochondrial defects, dysregulated metabolic processes and stalled cell division, but show that a single amino acid, aspartate, rescues multiple aspects of cell dysfunction, including proliferation. These results offer insights into the molecular effects of iron deficiency within the immune system.

## Results

### Transcriptomic and proteomic screens reveal potential nodes of dysfunction during CD8+ T cell iron deficiency

Iron is utilised by ~400 proteins, of which 204 are described to be expressed in T cells[7,14,15]. Iron-interacting proteins operate in a wide diversity of pathways ranging from mitochondrial metabolism to DNA synthesis[6,7], suggesting that iron deficiency likely has important and wide-acting effects on T cell biochemistry. However, the specific effects of iron scarcity are difficult to predict because of the interdependence of cellular processes. We therefore took an unbiased approach to initially define the impact of iron limitation on global CD8+ T cell transcription and translation. We employed RNA-sequencing (RNA-seq) and quantitative protein-mass spectrometry (protein-MS) of mouse OT-I CD8+ T cells activated in vitro across physiologically-titrated transferrin-iron concentrations (0.001–0.625 mg/mL holotransferrin, with total transferrin kept constant at 1.2 mg/mL) for 48 h (Fig. 1a). 0.625 mg/mL holotransferrin (15.6 μmol/L of iron) is representative of the levels of iron found in human sera (14–32 μmol/L of iron)[16]. Meanwhile, 0.001 mg/mL holotransferrin is much lower than levels found in plasma but may be reflective of the almost negligible levels of iron detected in lymphatic fluid[17].

In line with previous work showing that in vivo iron deprivation significantly reduces T cell expansion[5], in vitro iron limitation profoundly suppressed cellular proliferation, measured using cell trace violet (CTV; Fig. 1b, c). Iron deficiency also impaired expression of the activation marker, CD25, while having no effect on CD44 (Fig. 1d, e). Surface expression of the iron uptake receptor, TFR1/CD71, was increased in low iron conditions (Fig. 1f) consistent with iron limitation[18] while the amino acid transporter, CD98 (SLC3A2:SLC7A5 heterodimer; LAT1), which is induced post-T cell activation[19], was significantly decreased in low iron conditions (Fig. 1g). In line with previous reports[20,21], the cytokine, IL-2, was induced during iron deprivation (Fig. 1h). TNF was similarly induced with iron scarcity

(Fig. 1i) while IFN-γ was unchanged (Fig. 1j). Therefore, in low iron conditions, stimulated OT-I T cells acquire some though not all characteristics of activation on a per-cell basis, and because of a failure to proliferate, overall, there are fewer competent effector T cells when iron is restricted. Notably, the impaired proliferation and altered effector function observed under low iron conditions are not due to a failure to activate. Cells cultured in both low iron and iron-replete conditions show robust upregulation of CD25 expression and increased cell size (measured as forward scatter) relative to naïve cells maintained in IL-7 (Supplementary Fig. 1a, b). Further, cells in low iron display a transcriptional profile distinct from naïve cells, which is substantially more similar to cells in iron-replete conditions (Supplementary Fig. 1c).

Despite observing dramatic changes in cellular division during iron restriction, only 193 genes were identified as significantly differentially regulated between iron-replete and deficient conditions by RNA-seq (Supplementary Fig. 1d; Supplementary Data 1). However, clear segregation was still observed between the two iron concentrations by principal component analysis (PCA; Supplementary Fig. 1c). Genes identified by RNA-seq as induced during iron deficiency also displayed an enrichment of the activating histone marks, H3K4me3 and H3K27ac at transcription start sites (TSSs) in the low iron condition (e.g. *Cdkn1a*; Supplementary Fig. 1e, f) when analysed by ChIPmentation. Meanwhile, genes suppressed by iron deficiency at the RNA level showed depletion of these activating marks around their TSSs (e.g. *Asns*). The positive relationship between the transcriptionally associated histone modifications, H3K4me3 and H3K27ac, with transcription suggests that RNA expression changes observed during iron deficiency are likely due to transcriptional differences rather than alterations in RNA stability. In contrast, the iron uptake receptor, *Tfrc*, showed no observable difference in *Tfrc* promoter H3K4me3 or H3K27ac but a significant increase in mRNA and protein expression in low iron conditions (Supplementary Fig. 1f–h). These data are consistent with the canonical post-transcriptional upregulation of *Tfrc* via increased mRNA stability under iron limitation by the iron-response element (IRE)/iron-response protein (IRP) system[18].

Protein-MS analysis identified 116 differentially expressed proteins (Fig. 1k; Supplementary Data 2). As expected, TFR1 protein copy number increased as holotransferrin was depleted (Supplementary Fig. 1h). Despite the differential expression of 116 proteins, no differences were observed in overall protein mass or protein molecules per CD8+ T cell activated across a titration of holotransferrin conditions (Supplementary Fig. 1i, j), indicating that despite decreased cell division, iron-deficient CD8+ T cells do not accumulate extra protein.

5091 mRNAs/proteins were mutually detected by the RNA-seq and protein-MS with a significant positive correlation in the log2|FC| values (Fig. 1l). Of these, only 15 were significantly differentially regulated at both the mRNA and protein levels (Supplementary Fig. 1k). These modest expression changes indicate changes in protein function rather than expression may be the predominant driver of the profound suppressive effect of iron deficiency on CD8+ T cell proliferation. Of note, 7 of the 15 mutually differentially expressed genes encoded proteins related to metabolism, including amino acid transporters (*Slc1a4*, *Slc7a3*) or enzymes (*Dglucy*, *Ephx1*, *Asns*, *Cth*, *Pck2*), indicating changes in cellular metabolism may underpin this functional alteration.

In addition, gene set enrichment analysis (GSEA), conducted on both RNA-seq and protein-MS datasets, also indicated altered expression of genes attributed to the p53 pathway (Fig. 1m, Supplementary Fig. 1l). The p53 pathway coordinates responses to cellular stress such as DNA damage, hypoxia and nutrient deprivation, with downstream effects including apoptosis, DNA repair and cell cycle arrest[22]. p53 has also been shown to regulate cellular metabolism with

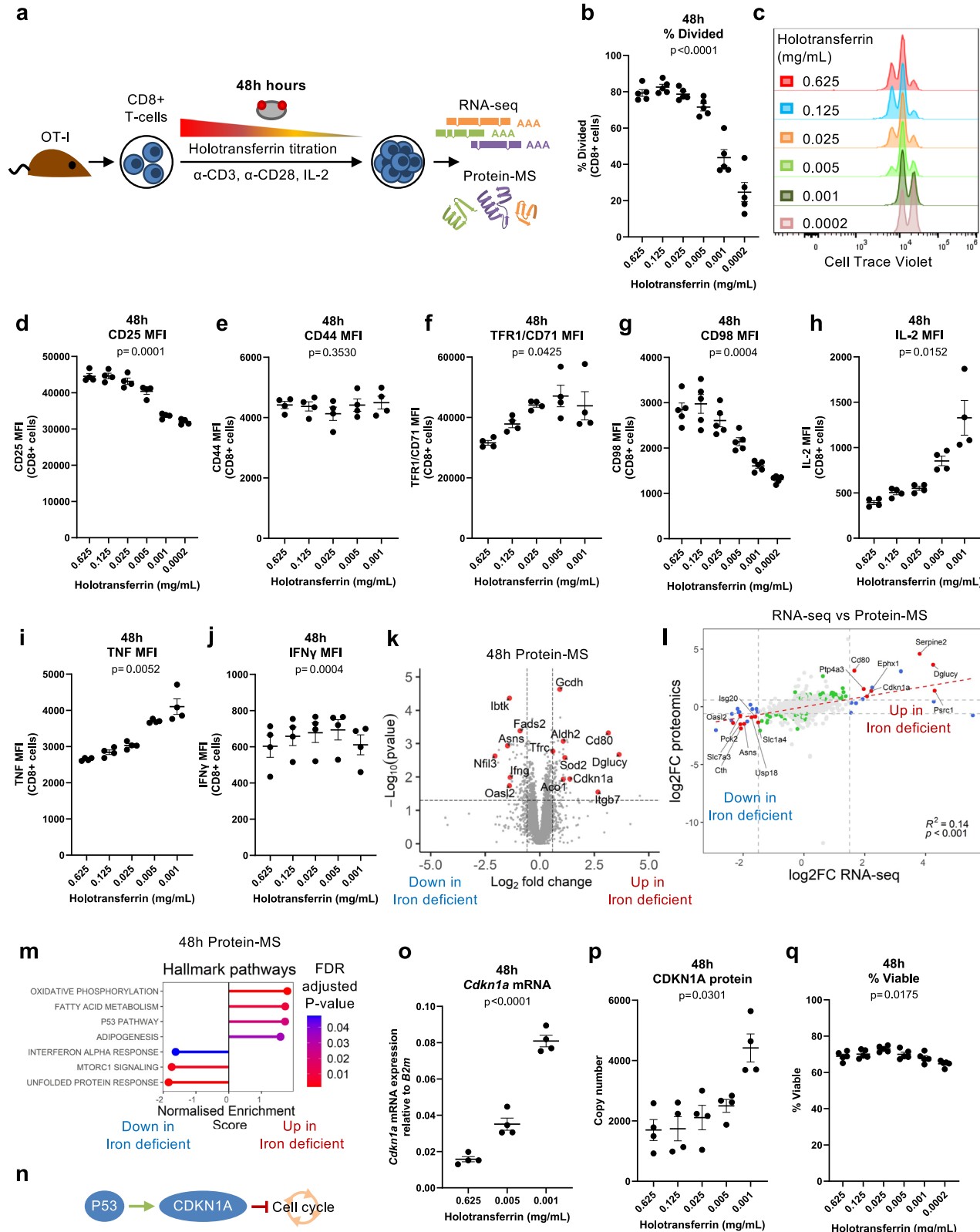

effects including glycolytic suppression and mitochondrial maintenance[23]. Altered activity of genes involved in the p53 pathway may therefore also contribute to the titratable impairment of cellular proliferation in low iron conditions (Fig. 1b, c). Consistent with this, expression of *Cdkn1a* (also known as *p21*, *Cip1* and *Waf1*), a p53 target and suppressor of the G1-S phase cell cycle transition was induced by iron deprivation at 24 h, prior to the first cell division and continued to

be upregulated as CD8+ T cells entered their proliferative phase at 48 h (Fig. 1n–p, Supplementary Fig. 1m). However, while cell division halted, cells remained viable at 48 h (Fig. 1q). Notably, *Cdkn1a* is induced by iron chelation in cell lines[24]. This data suggests that induction of *Cdkn1a* expression, potentially via the p53 signalling pathway, may contribute to drive cell cycle suppression, but iron deprivation does not cause cell death at this time point.

**Fig. 1 | Iron deficiency induces changes to metabolic processes at the RNA and protein levels. a** CD8+ OT-I T cells isolated from mice were activated with 5 μg/mL plate-bound α-CD3, 1 μg/mL α-CD28 and 50 U/mL IL-2 for 48 h in a titration of holotransferrin conditions. Naïve T cells were collected on day 0. Where comparisons between high and low iron conditions are made, the holotransferrin concentrations used are 0.625 (high) and 0.001 (low) mg/mL. **b, c** Division assessed using cell trace violet (CTV), $n = 5$. **d** CD25, **e** CD44, **f** TFR1/CD71, **g** CD98, **h** IL-2, **i** TNF and **j** IFN-γ mean fluorescence intensity (MFI), $n = 4$ except for (**g**) where $n = 5$. **k** Volcano plot with the significance thresholds of $\log_2|$fold change (FC)$| > 0.585$, $p$ value $< 0.05$, $n = 4$. No correction for multiple testing was applied. **l** Correlation plot of the $\log_2|$FC$|$ between high and low iron conditions by RNA-seq and protein-mass spectrometry (protein-MS), $n = 4$. **m** Hallmark gene set enrichment analysis (GSEA) for the protein-MS, $n = 4$. A Benjamini-Hochberg adjustment was used to correct for multiple testing. **n** p53 induces CDKN1A expression which inhibits the G1-S phase transition. *Cdkn1a* **o** mRNA and **p** protein expression and **q** percentage viable cells, $n = 4$. Data are mean ± standard error of the mean (SEM), where each datapoint per condition denotes cells from independent donor mice. Statistics are: (**b, d–j, o–q**) sampled matched one-way ANOVAs with the Geisser-Greenhouse correction; **l** two-sided Pearson correlation $R^2$ value. Source data are provided as a Source Data file.

## Low iron availability induces a response distinctive from either hypoxia or iron chelation

p53 induction during iron chelation has been described in cell lines, macrophages and CD4+ T cells[25–28] and is understood to occur via HIF1α stabilisation due to impaired activity of their regulators, the iron and oxygen-dependent PHD proteins[28]. We therefore tested whether HIF1α was similarly stabilised in cells deprived of iron more physiologically, via decreased availability of transferrin-bound iron. While the iron chelator deferiprone (DFP) clearly induced HIF1α stabilisation in CD8+ T cells, low iron conditions did not alter HIF1α levels relative to iron-replete controls (Supplementary Fig. 2a, b). Moreover, the proteomic profile of iron-deficient CD8+ T cells bears little resemblance to the proteomes of 5 day activated cytotoxic CD8+ T cells exposed to hypoxia for 24 h as reported by Ross et al.[29] (Supplementary Fig. 2c). Both hypoxia and iron chelation have also been shown to induce glycolytic genes[25,29], but this was not observed in our cells in low iron media.

Given the discrepancy between how HIF1α responded in CD8+ T cells under iron chelation and iron deficiency, we were interested to understand whether the difference in cellular response extended beyond HIF1α. We processed raw RNA-seq data from Wang et al.[30] (not analysed in their manuscript) of Th1 polarised CD4+ T cells treated with CPX iron chelator or dimethyl sulfoxide control for 4 h. CPX-treated cells showed differential expression of 3523 genes, which is ~18-fold higher than the 193 differentially expressed genes identified in our model of low iron media (Supplementary Fig. 2d). CPX also failed to upregulate *Tfrc* (Supplementary Fig. 2e), suggesting that the IRP-IRE iron signalling system is no longer intact. While CPX iron chelation suppressed pathways associated with cell cycle (G2M checkpoint) and metabolic reprogramming (MYC targets; Supplementary Fig. 2f), similar to iron deficiency, only 45 genes were found to be mutually differentially expressed (Supplementary Fig. 2g). The extreme changes in transcriptional profile observed with only 4 h of CPX treatment relative to our 48-h culture model in low iron media and the failure to upregulate *Tfrc* suggest that the biochemical impacts of chelators can be non-physiological, especially if given at high doses.

## Low iron alters the CD8+ T cell mitochondrial proteome

We observed decreased activity of the key upstream metabolic regulator, mTOR, measured as phosphorylation of its target S6 (pS6; Fig. 2a–c) and downregulation of the MYC and mTORC1 signalling pathways under iron scarcity (Fig. 1m, Supplementary Fig. 1l). Decreased mTORC1 is consistent with lack of upregulation of glycolysis genes, reduced CD98 expression (Fig. 1g), and may also relate to lack of total protein accumulation in non-proliferating iron-deficient cells (Supplementary Fig. 1i, j). However, unexpectedly, we observed upregulation of proteins involved in mitochondrial processes, including OXPHOS and fatty acid metabolism (Fig. 1m). The enrichment of the fatty acid signature was driven largely by increases in β-oxidation proteins involved in fatty acid and branched chain amino acid breakdown rather than fatty acid synthesis (Fig. 2d). ETC proteins involved in the highly iron-dependent respiratory complexes I (CI), CIII and CIV were also upregulated. Consistent with increased abundance

of these mitochondrial proteins, mitochondrial mass, measured with Mitotracker green (MTG), was also elevated in low iron conditions (Fig. 2e). Of note, by selecting for mitochondrially localised proteins using the MitoCarta3.0 gene set[31], we observed improved segregation within the first two principal components of CD8+ T cells cultured in different iron concentrations compared to when all proteins were analysed (Fig. 2f, g), indicating mitochondrial proteins are disproportionately influenced by iron availability relative to all proteins detected.

Mitochondria are enriched for iron-interacting proteins, with 7% of mitochondrial proteins classified as iron-interacting compared to 2% cell-wide[7]. Aconitase 2 (ACO2) and SDH of the TCA cycle and CI–CIV of the ETC require iron cofactors for function[32]. Mitochondria are also home to haem and iron–sulphur (Fe–S) cluster synthesis pathways[32]. Analysis of proteomics data from Howden et al.[15] revealed upregulation of haem and Fe-S cluster synthesis proteins in T cells 24 h and 6 days post-activation (Supplementary Fig. 3a–c). The heavy reliance of mitochondrial function on iron in combination with the upregulation of mitochondrial proteins suggests that mitochondrial function may be disrupted during iron scarcity.

## Iron deficiency impairs CD8+ T cell mitochondrial function

To directly interrogate CD8+ T cell mitochondrial function under iron deprivation, we first assessed a metric of mitochondrial health, specifically levels of superoxide ($O_2^{\cdot-}$)—a key exemplar of mROS species. As available iron declined, mROS levels increased in CD8+ T cells (Fig. 2h, i). CI, CII and CIII of the ETC, which all require iron for electron transfer[32] are key mROS producers[33]. Given the observed increase in ETC proteins (Fig. 2d), but lower availability of iron during iron restriction, we propose an imbalance in ETC proteins to iron cofactors may impair efficient electron transfer, resulting in increased mROS generation. Consistent with this, iron deprivation decreased the mitochondrial membrane potential (Fig. 2j, Supplementary Fig. 4a) and was previously shown to suppress mitochondrial ATP generation[5]. Upregulation of the mitochondrial superoxide detoxifying protein, SOD2, was also observed under iron scarcity (Fig. 2k), indicating compensatory mechanisms to suppress mROS, which at the lowest iron concentrations still fail to control them.

Taken together with the alterations in the mitochondrial proteome, these data indicate that iron-deprived CD8+ T cells may have defective mitochondrial metabolism. To directly assess this, we next performed stable isotope-based tracing coupled to metabolite-mass spectrometry (metabolite-MS). CD8+ T cells were activated for 24 h and then incubated with $^{13}C_6$-glucose or $^{13}C_5$-glutamine for a further 24 h under iron-replete or -deficient conditions. Here, minimal changes in overall abundance of the glycolytic metabolites, pyruvate and lactate, or the amino acids which may be derived from glycolytic intermediates or imported (alanine, serine and glycine) were observed (Fig. 3a). Furthermore, there were no significant changes in the abundance of total lactate or the fraction of lactate labelled from $^{13}C_6$-glucose in the supernatants of iron-deprived CD8+ T cells (Supplementary Fig. 4b, c) together suggesting glycolytic activity was preserved under low iron conditions. This is consistent with minimal changes in HIF1α activity (Supplementary Fig. 2a–c) and in agreement

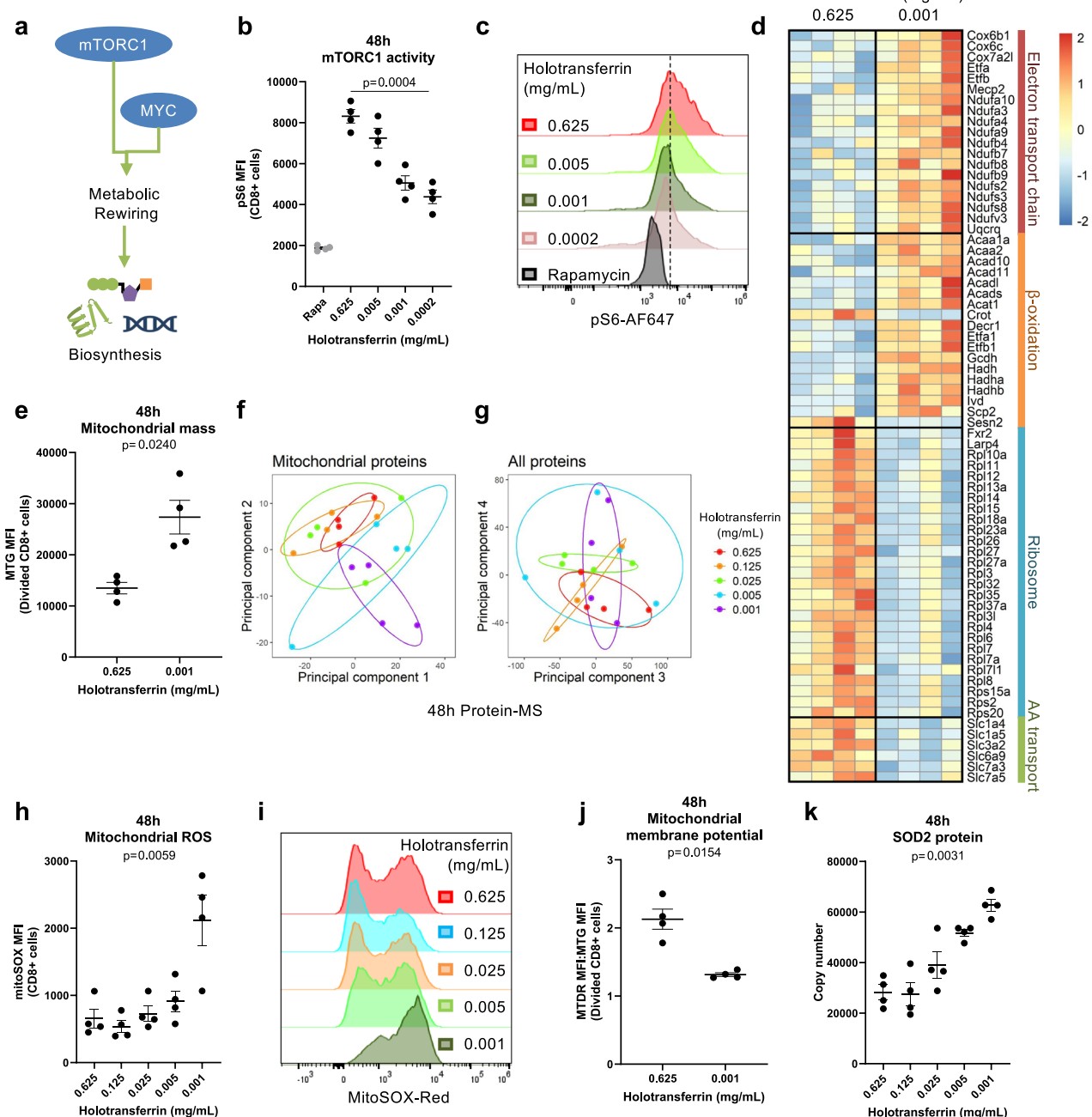

**Fig. 2 | Iron deprivation induces changes to the mitochondrial proteome, mitochondrial reactive oxygen species (mROS) production and loss of mitochondrial membrane potential.** CD8+ T cells were activated as described in Fig. 1a. **a** mTORC1 and MYC are metabolic regulators that enable biosynthesis downstream of TCR stimulation. **b**, **c** mTORC1 activity measured via its downstream target, phospho-S6 (pS6), $n = 4$. Controls were treated overnight with rapamycin (rapa; 1 μM). **d** Heatmap of proteins in selected metabolic pathways defined using GO terms where the $p$ value < 0.05, $n = 4$. Electron transport chain = GO_RESPIRATORY_ELECTRON_TRANSPORT_CHAIN, β-oxidation = GO_FATTY_ACID_BETA_OXIDATION, ribosome = GO_CYTOSOLIC_RIBOSOME, amino acid (AA) transport = GO_AMINO_ACID_IMPORT. **e** Mitochondrial membrane mass measured using Mitotracker green (MTG) MFI, $n = 4$. PCA of the protein-MS **f** given prior selection for proteins in the MitoCarta3.0 gene set and **g** of all proteins, $n = 4$. **h**, **i** mROS MFI measured using MitoSOX red, $n = 4$. **j** Mitochondrial membrane potential calculated as the ratio of Mitotracker deep red (MTDR) to Mitotracker green (MTG), $n = 4$. **k** SOD2 protein expression by protein-MS, $n = 4$. Data are mean ± SEM, where each datapoint per condition denotes cells from independent donor mice. Statistics are: **b**, **h**, **k** matched one-way ANOVAs with the Geisser-Greenhouse correction; **e**, **j** matched two-tailed $t$-test. Source data are provided as a Source Data file.

with glycolysis being an iron-independent pathway (none of the enzymes involved in glycolysis require iron).

In these experiments we also noted that while ~20% labelling of $^{13}C6$-glucose into the TCA cycle was observed, $^{13}C5$-glutamine was much more readily incorporated into the TCA cycle, labelling 60-80% of TCA cycle metabolites (Supplementary Fig. 4d). The larger

incorporation of glutamine relative to glucose into the TCA cycle is consistent with reports indicating substantial reduction of glucose-derived pyruvate to lactate in activated T cells, alongside increased glutamine anaplerosis into the TCA cycle[34].

ACO2 is an iron-dependent enzyme in the TCA cycle, which converts citrate to isocitrate. Isocitrate dehydrogenase 2 (IDH2) then

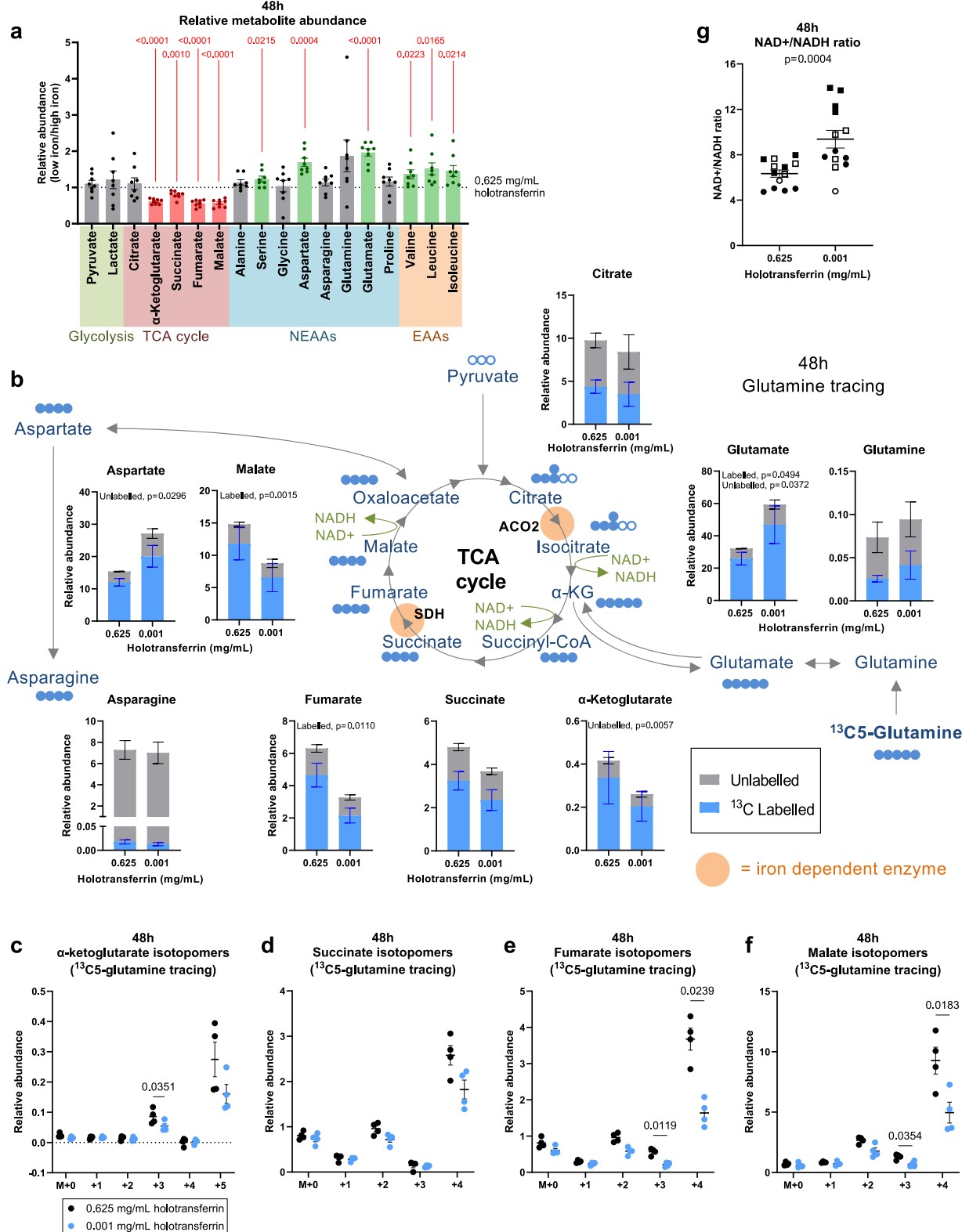

converts isocitrate to α-ketoglutarate (α-KG). In agreement with reduced TCA cycle activity, decreased abundance of α-KG was observed in low iron cells (Fig. 3a, b). Decreased α-KG M + 3 isotopomers from ¹³C5-glutamine and a trend towards reduced α-KG M + 2 isotopomers from ¹³C6-glucose support reduced ACO2 and/or IDH2 activity (Fig. 3c, Supplementary Fig. 4e–g). However, since α-KG is also produced from glutamate by the activity of glutamate

dehydrogenase (GDH), decreased α-KG may also reflect decreased activity of this enzyme. Indeed, decreased relative abundance of α-KG M + 5 mass isotopomers from ¹³C5-glutamine indicates a trend towards decreased GDH activity under low iron conditions (Fig. 3c, Supplementary Fig. 4e). Consistent with this, ¹³C-labelled glutamate accumulated in low iron conditions following ¹³C5-glutamine tracing (Fig. 3b).

**Fig. 3 | Iron scarcity impairs tricarboxylic acid (TCA) cycle activity at the iron-dependent enzymes ACO2 and SDH.** CD8+ T cells were activated as described in Fig. 1a. For tracing experiments, T cells were activated in standard media for 24 h and then incubated in media containing $^{13}$C6-glucose or $^{13}$C5-glutamine for a further 24 h. **a** Relative metabolite abundance from T cells cultured in low iron (0.001 mg/mL holotransferrin) versus high iron (0.625 mg/mL holotransferrin), normalised to spiked in glutaric acid. Pooled relative total abundances from the $^{13}$C6-glucose and $^{13}$C5-glutamine experiments, n = 8. NEAA non-essential amino acids, EAA essential amino acids. *P* values are shown in red. **b** $^{13}$C5-glutamine tracing, n = 4. Relative abundance of labelled and unlabelled metabolites calculated as the fraction labelled multiplied by the raw glutaric acid normalised abundance. Blue-filled circles indicate carbon atoms labelled from $^{13}$C5-glutamine. Empty circles indicate unlabelled carbon atoms. Relative abundance of **c** α-KG, **d** succinate, **e** fumarate and **f** malate mass isotopomers from $^{13}$C-glutamine tracing calculated as the fraction labelled multiplied by the raw glutaric acid normalised abundance, n = 4. **g** NAD+/NADH ratio. Data from independent experiments denoted by different symbols, n = 13. Data are mean ± SEM, where each datapoint per condition denotes cells from independent donor mice. Statistics are: **a** matched two-way ANOVA with the Geisser-Greenhouse correction and the Fisher's least significant difference (LSD) test for multiple comparisons; **b**–**f** matched two-way ANOVAs with the Geisser-Greenhouse correction and the Sidak correction for multiple comparisons; **g** matched two-tailed *t*-test. Source data are provided as a Source Data file.

The TCA cycle metabolite succinate lies upstream of the second iron-dependent TCA cycle enzyme (SDH). Consistently, we observed that while succinate abundance was depressed in iron limiting conditions (−23%; Fig. 3a, b), it was relatively accumulated compared to the downstream metabolites, fumarate (−47%) and malate (−41%), suggesting suppressed progression from succinate to fumarate (Fig. 3a, b). Decreased M + 4 $^{13}$C-labelling from $^{13}$C5-glutamine into fumarate and malate also supports this (Fig. 3d–f).

Decreased oxidative TCA cycle progression, due to limited glutamine anaplerosis and ACO2 and SDH activity, agrees with diminished NAD+ reduction and consequent increases in the NAD+/NADH ratio (Fig. 3g, Supplementary Fig. 4h), also observed under iron scarcity. Since NADH is a positive allosteric regulator of GDH, this may also partly explain the indicated decrease in GDH activity[35].

### Iron depletion alters H3K27 methylation, but proliferation is not rescued by α-KG supplementation

α-KG levels were decreased by ~40% in iron-deficient CD8+ T cells (Figs. 3a, b and 4a). α-KG is a substrate for many dioxygenases, including the histone lysine demethylases (KDMs)[36] (Fig. 4b), and alterations in T cell metabolism have been shown to alter KDM activity and T cell fate[37]. Crucially, most KDMs require an iron catalytic core to mediate hydroxylation of histone methyl groups using α-KG and oxygen as substrates[36]. The unstable methyl-hydroxy intermediate spontaneously dissociates, leaving a demethylated product[38,39]. This dual dependency of KDMs for α-KG and iron means that a decrease of α-KG availability under iron limitation may exert a double hit on KDM activity, with potential impacts on the appropriate chromatin restructuring necessary for CD8+ T cell differentiation. The abundance of KDMs substantially increases upon T cell activation, with KDM6B being the second most upregulated KDM in CD8+ T cells[14,15]. KDM6B is responsible for the removal of the repressive histone mark, H3K27me3, a process critical for T cell differentiation and effector function acquisition[40,41]. Consistent with decreased activity under iron limitation, CD8+ T cells demonstrated a titratable failure to remove H3K27me3 relative to iron-replete controls, with H3K27me3 levels remaining almost as high as IL-7-treated, naïve-like cells (Fig. 4c, d, Supplementary 5a, b). Using ChIPmentation, we confirmed that H3K27me3 levels were significantly elevated at TSSs in iron-deficient CD8+ T cells relative to iron-replete (Fig. 4e, f). The genome-wide increase in H3K27me3 during iron deficiency agrees with the concept that iron deprivation generally impairs the activity of KDM6 enzymes during T cell activation. It is also possible that increased methylation by methylases could contribute to the H3K27me3 accumulation phenotype. These findings therefore indicate that altered CD8+ T cell metabolic activity under low iron may have direct implications for their epigenetic status and differentiation capacity. While we hypothesise that KDM6 impairment occurs due to iron deficiency coupled to suppressed α-KG production, KDM6 activity may be impaired by other mechanisms. For instance, ROS, which are induced by iron deficiency (Fig. 2h), have been shown to inhibit other iron-dependent, oxygen-dependent and α-KG-dependent dioxygenases such as the PHD proteins, which regulate HIF1α[42]. Of note, however, direct

supplementation with cell-permeable dimethyl-α-KG failed to rescue H3K27me3 levels and cellular proliferation (Supplementary Fig. 5c, d) in iron-deprived T cells. This is consistent with KDM enzymes requiring all of iron, oxygen and α-KG. Thus, even if α-KG is no longer limiting, KDM enzymes will remain dysfunctional in the absence of sufficient iron cofactors.

### Iron depletion suppresses nucleotide synthesis from aspartate

Aspartate is produced downstream of the TCA cycle metabolite oxaloacetate and critically supports cellular proliferation through de novo purine and pyrimidine synthesis[43–45]. In cancer cells, ETC inhibition or iron chelation impairs aspartate synthesis, resulting in suppressed proliferation of these immortalised cells[46–48]. However, despite TCA cycle inhibition, aspartate was unexpectedly higher in iron-deficient cells, with the majority derived from glutamine (M + 4 labelled; Figs. 3a, b and 5a, Supplementary Fig. 6a).

To understand whether nucleotide synthesis was altered downstream of decreased TCA activity in iron-depleted CD8+ T cells, we measured the abundance of nucleotides and their upstream precursors by LC-MS, observing that AICAR, a metabolite which lies two steps downstream of aspartate incorporation into purine synthesis[49] was substantially decreased during iron limitation (Fig. 5b, c). PPAT, the initiating enzyme of purine synthesis, which has been predicted to be iron-dependent[7] was also reduced in iron scarcity (Fig. 5d). However, whether PPAT enzymatic activity is reduced is unclear. Carbamoyl-aspartate and orotate, which lie downstream of aspartate incorporation into pyrimidine synthesis, were similarly depleted (Fig. 5e, f). Consistent with these observations, certain nucleotides and dNTPs were decreased during iron deficiency (Supplementary Fig. 6b–d), albeit to a lesser extent, which may be explained by their decreased usage under suppressed CD8 + T cell proliferation and associated DNA synthesis (Fig. 1b, c) or increased salvage.

Together, these data indicate that nucleotide synthesis from aspartate is limited in iron-deficient CD8+ T cells, despite overall cellular abundance of aspartate being increased rather than decreased (Figs. 3a, b and 5a). This aspartate accumulation could be somewhat explained by increased production by an alternative pathway, for instance, pyruvate anaplerosis to oxaloacetate via pyruvate carboxylase (PC) and a partial reversal of TCA cycling (Fig. 5g)[50]. Carbon labelling from $^{13}$C6-glucose through the oxidative (canonical forward) TCA cycle via pyruvate dehydrogenase (PDH) results in entry of two $^{13}$C atoms into TCA cycle metabolites (M + 2 mass isotopomers)[50]. Meanwhile, entry of carbon from $^{13}$C6-glucose to the reductive (reverse) TCA cycle via PC produces M + 3 mass isotopomers. During iron deficiency, significant increases in $^{13}$C6-glucose labelling into M + 3 TCA cycle metabolites, including fumarate and malate, were observed, indicative of increased PC contribution to these metabolites and reductive TCA cycling from oxaloacetate to fumarate (Fig. 5h). M + 3 and M + 5 citrate isotopomers are also likely derived from PC routed $^{13}$C6-glucose atoms (Fig. 5h, Supplementary Fig. 6e). PC activity, evaluated as (malate M + 3−succinate M + 3)/pyruvate M + 3, as utilised by Elia et al.[51], was also increased (Fig. 5i). PC protein also trended towards increased expression in low iron conditions (Fig. 5j) while PCK2, which mediates

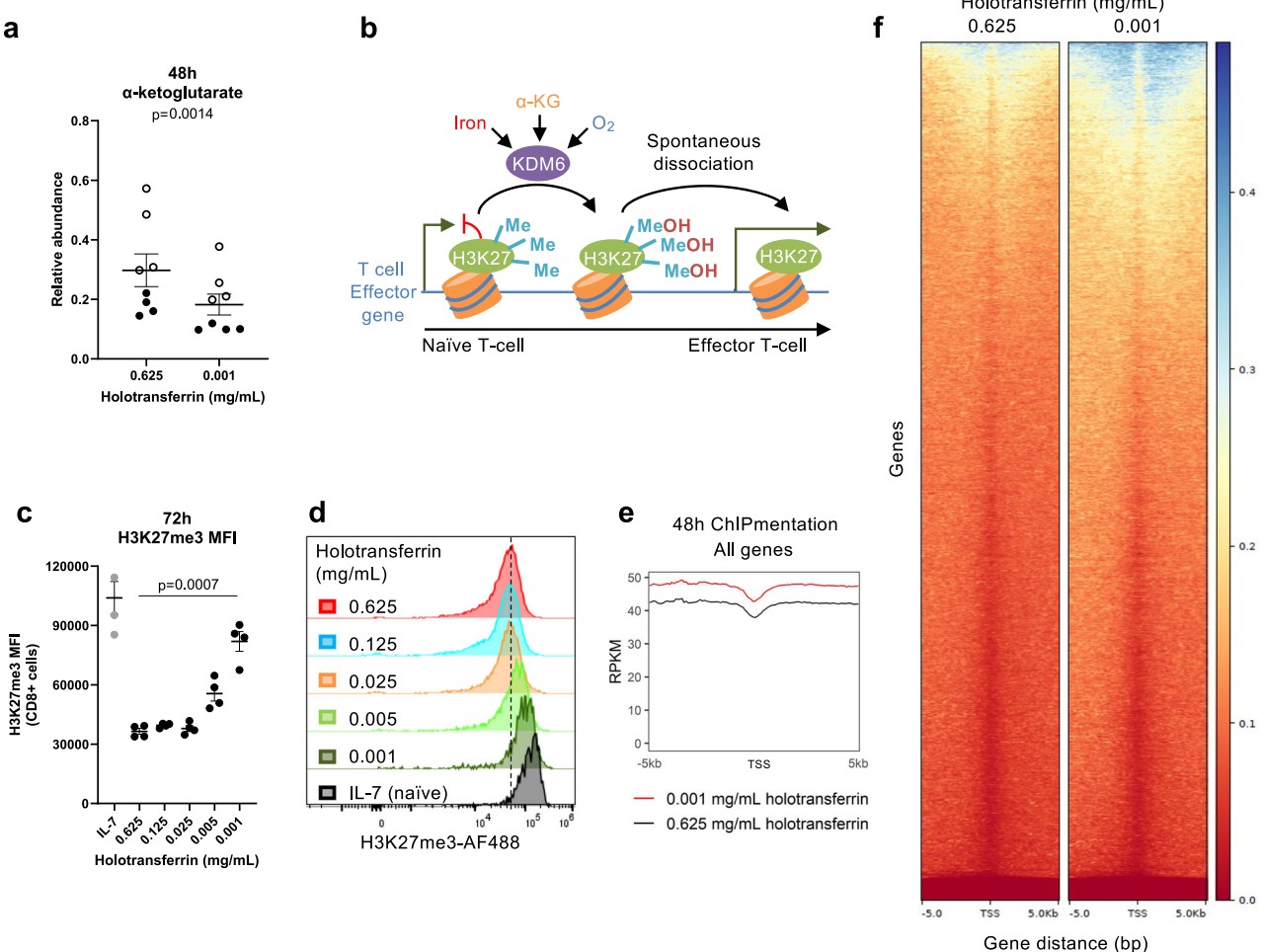

**Fig. 4 | Iron deficiency permits the accumulation of H3K27me3 in CD8+ T cells.** CD8+ T cells were activated as described in Fig. 1a. **a** Relative abundance of α-KG. Data from independent experiments denoted by different symbols, $n = 8$. **b** Lysine demethylase (KDM) enzymes use iron cofactors and α-KG and oxygen substrates to mediate the hydroxylation of methyl groups, which spontaneously dissociate to leave a demethylated histone. KDM6 enzymes remove the repressive histone mark, H3K27me3, from effector gene loci upon T cell activation. **c, d** H3K27me3 MFI. Naïve-like control cells were cultured in IL-7 (5 ng/mL), $n = 4$. **e, f** H3K27me3 enrichment at transcription start sites (TSSs) of all genes assessed by ChIPmentation, $n = 3$. RPKM reads per kilobase per million mapped reads, bp basepairs. Data are mean ± SEM, where each datapoint per condition denotes cells from independent donor mice. Histograms are normalised to the mode. Statistics are: **a** paired two-tailed $t$-test; **c** matched one-way ANOVA with the Geisser-Greenhouse correction. Source data are provided as a Source Data file.

the reverse reaction, converting oxaloacetate to the glycolytic intermediate phosphoenolpyruvate (PEP) was suppressed (Fig. 5k) suggesting oxaloacetate production is favoured in low iron conditions. In line with increased PC usage, pyruvate supplementation provided a proliferative advantage to iron-depleted cells (Fig. 5l). A partial reversal of TCA cycle activity is also in agreement with the observed increase in the NAD+/NADH ratio (Fig. 3g, Supplementary Fig. 4h). This suggests that under iron limiting conditions, CD8+ T cells may utilise PC and a partial reversal of the TCA cycle to replenish the depleted metabolites fumarate and malate, circumventing the use of the iron-dependent enzymes, ACO2 and SDH. Alternatively, increased PC activity may be a consequence of a futile attempt to increase nucleotide production via aspartate. Aspartate accumulation under iron depletion could also be explained by decreased usage, as shown for nucleotide synthesis above. Aspartate is also used for the synthesis of asparagine, however at 48 h post-activation, CD8+ T cells in both iron-deficient and -replete conditions had not synthesised asparagine, indicated by the absence of [13]C-labelling from [13]C5-glutamine (Fig. 3b, Supplementary Fig. 6f). This agrees with previous reports that CD8+ T cells lack asparagine producing capacity (asparagine synthetase expression) at 48 h post-activation, but gain this later into their activation and differentiation[45].

Accordingly, asparagine supplementation provided no proliferative benefit (Supplementary Fig. 6g). Taken together, the increased abundance of aspartate alongside decreased nucleotide synthesis indicates that aspartate usage by nucleotide synthesis is suppressed under iron restriction.

### Increased nucleotide availability provides resistance to iron deficiency in CD8+ T cells
To interrogate whether nucleotide abundance limits CD8+ T cell proliferation under iron deficiency, we first tested whether genetically increasing nucleotide abundance could rescue proliferation. Cellular nucleotide balance is maintained via the activity of iron-dependent ribonucleotide reductase (RNR) and SAMHD1[52] (Supplementary Fig. 7a). While RNR enables production of dNTPs, SAMHD1 degrades dNTPs[52]. Consistently, *Samhd1* deletion (KO) results in dNTP accumulation in lung fibroblasts[53] and bone marrow-derived DCs[54] and is assumed to operate similarly in CD8+ T cells. We isolated CD8+ T cells from *Samhd1*-KO or wild-type littermates and activated them as described above. *Samhd1*-KO CD8+ T cells were less sensitive to iron scarcity in terms of a block on proliferation relative to wild-type cells (Supplementary Fig. 7b). This effect was modest, likely explained by

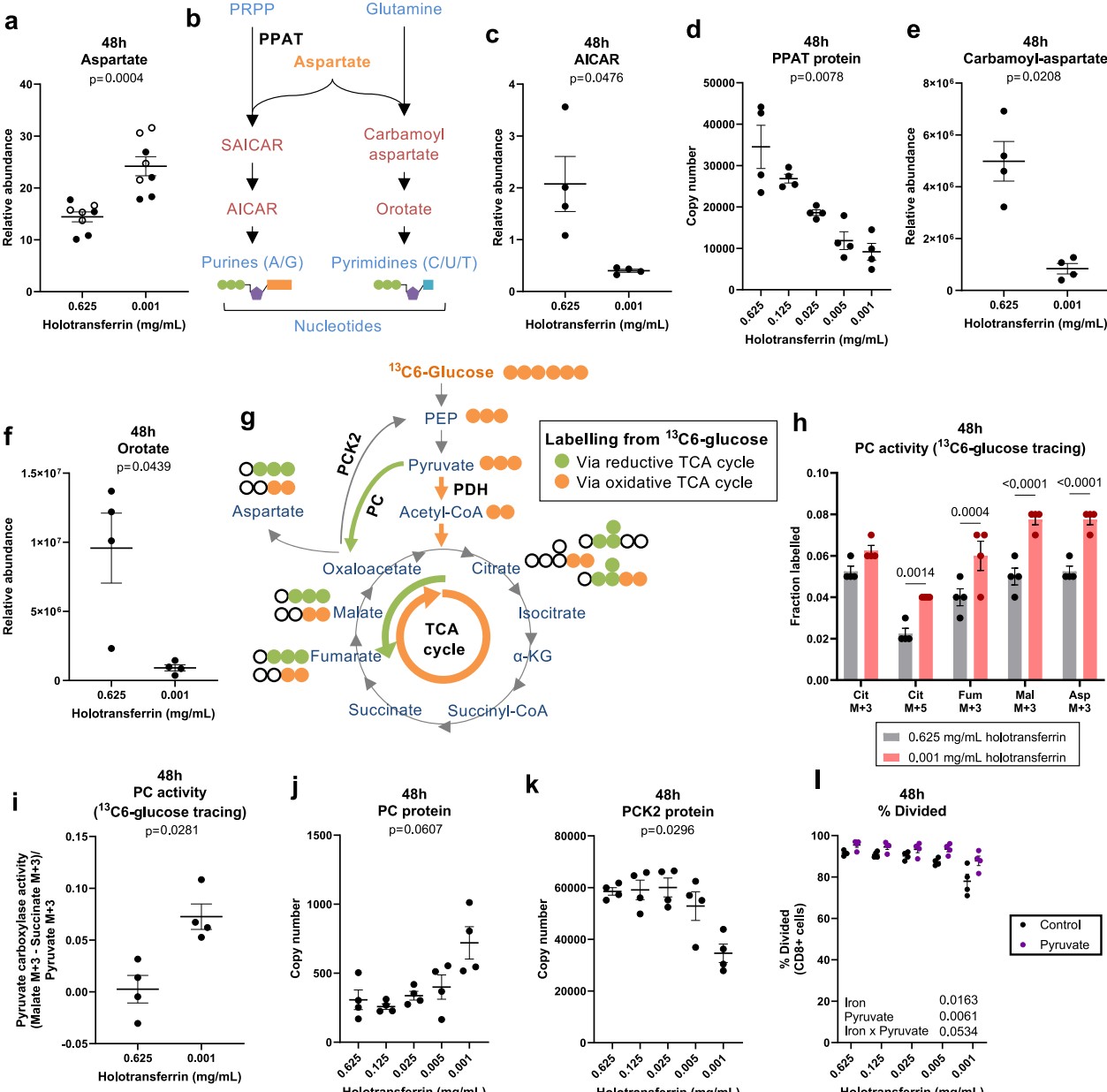

**Fig. 5 | Iron scarcity suppresses nucleotide synthesis downstream of aspartate incorporation.** CD8+ T cells were activated as described in Fig. 1a. For the ¹³C6-glucose tracing experiments, T cells were activated for 24 h and then incubated in media containing ¹³C6-glucose for a further 24 h. **a** Relative abundance of aspartate normalised to a glutaric acid spike in. Data from independent experiments denoted by different symbols, $n = 8$. **b** Aspartate is incorporated into purine and pyrimidine nucleotides. Relative abundance of **c** AICAR, $n = 4$. AICAR was normalised to spike in ¹⁵N-dT. **d** PPAT protein via protein-MS, $n = 4$. Relative abundance of **e** carbamoyl-aspartate and **f** orotate, $n = 4$. Carbamoyl-aspartate and orotate were normalised to be spiked in glutaric acid. **g** Schematic of ¹³C6-glucose tracing. Orange and green circles indicate ¹³C-labelled atoms. Orange circles show labelling expected from oxidative TCA cycling via PDH. Green circles indicate labelling from reductive TCA cycling via PC. PC activity measured via the **h** fractional labelling into heavy labelled metabolites expected from reductive TCA cycling and via **i** the ratio of (Malate M3–Succinate M3)/Pyruvate M3, $n = 4$. **j** PC protein expression via protein-MS, $n = 4$. **k** PCK2 protein expression via protein-MS, $n = 4$. **l** Division measured using cell trace violet (CTV) with or without pyruvate (10 mM), $n = 4$. Data are mean ± SEM, where each datapoint per condition denotes cells from independent mice. Statistics are: **a, c, e, f, i** paired two-tailed *t*-tests; **d, j, k** one-way ANOVAs with the Geisser-Greenhouse correction; **h** matched two-way ANOVA with the Sidak correction for multiple comparisons; **l** matched two-way ANOVA with the Geisser-Greenhouse correction. Source data are provided as a Source Data file.

the fact that SAMHD1-KO prevents dNTP breakdown but cannot rescue impaired nucleotide production or iron-dependent RNR activity. SAMHD1-KO CD8+ T cells showed appropriate upregulation of *Tfrc* expression in low iron concentrations (Supplementary Fig. 7c, d) but failed to rescue CD25 or perforin expression (Supplementary Fig. 7e, f). Notably, SAMHD1-KO cells showed comparable expression of *Cdkn1a* (Supplementary Fig. 7g), indicating that increased nucleotide pools provide a proliferative advantage despite elevated *Cdkn1a* expression.

**Aspartate supplementation rescues iron-deficient CD8+ T cells**
For aspartate to be utilised for nucleotide synthesis, it must be in the cytosol. Thus, an explanation for decreased aspartate usage could be inappropriate retention of aspartate in the mitochondria. SLC25A12/13 transports aspartate from the mitochondria to the cytosol in exchange for glutamate[55] (Fig. 6a), but requires a proton gradient to mediate transport[55], which is decreased in iron-depleted CD8+ T cells. (Fig. 2j). Therefore, it is possible that a reduction in ETC chain activity due to

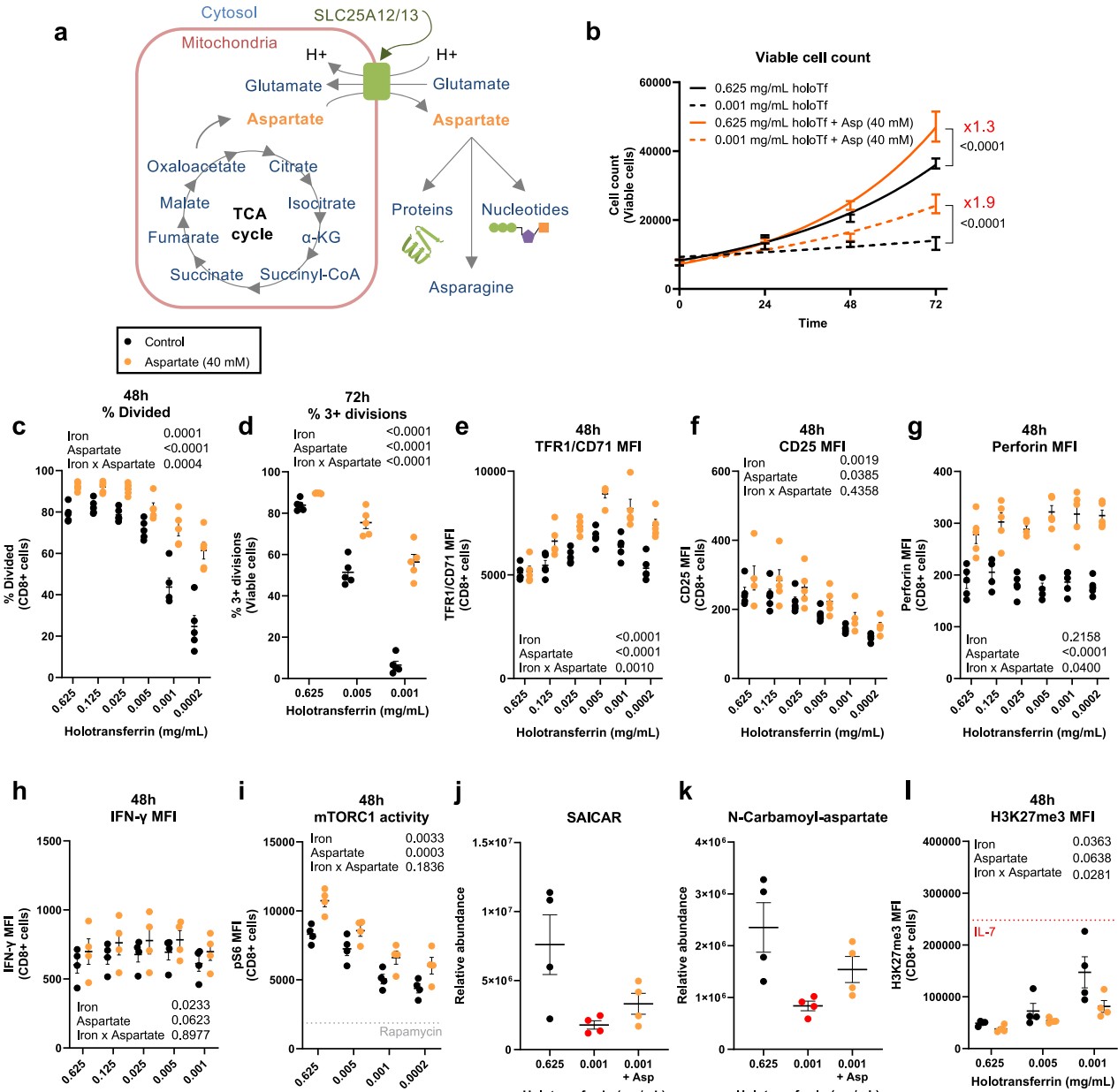

**Fig. 6 | Aspartate increases the carrying capacity of iron-deprived CD8+ T cell cultures.** CD8+ T cells were activated as described in Fig. 1a with or without aspartate (40 mM). **a** Aspartate is synthesised in the mitochondria but must be transported into the cytosol by the proton-dependent transporter, SLC25A12/13, for downstream metabolism. **b** Viable cell counts, $n = 5$. **c** Percentage of divided cells at 48 h and **d** percentage of cells undergoing 3+ divisions at 72 h assessed using cell trace violet (CTV), $n = 5$. **e** TFR1/CD71, **f** CD25 and **g** perforin MFI, $n = 5$. **h** IFN-γ MFI, $n = 4$. **i** mTORC1 activity measured via phospho-S6 (pS6), $n = 4$. **j** SAICAR and **k** N-carbamoyl-aspartate relative abundance normalised to spiked-in glutaric acid, $n = 4$. **l** H3K27me3 MFI, $n = 4$. Naïve-like control cells were cultured in IL-7 (5 ng/mL). Data are mean ± SEM, where each datapoint per condition denotes cells from independent donor mice. Statistics are: **b** non-linear regressions using exponential growth equations with an extra sum-of-squares $F$ test applied for either high or low holotransferrin concentrations between aspartate-treated and untreated; **c–i**, **l** two-way ANOVAs with the Geisser-Greenhouse correction; **j**, **k** one-way ANOVAs with the Geisser-Greenhouse correction. Source data are provided as a Source Data file.

iron deficiency may impair the mitochondrial proton gradient, in turn inhibiting SLC25A12/13 and sequestering aspartate in the mitochondria where it cannot be used for nucleotide synthesis.

We directly assessed whether supplementation of aspartate into cell culture media, such that it can be taken up directly into the cytosol, can rescue CD8+ T cell proliferation. CD8+ T cells were activated in vitro as described in Fig. 1a in the presence or absence of aspartate (40 mM). While CD8+ T cells in low iron conditions with no added aspartate showed almost no population expansion over 72 h of culture, the addition of aspartate increased the carrying capacity of the low

iron culture by ~$10^4$ additional cells (Fig. 6b). When assessing division at 48 h using CTV, aspartate increased the percentage of divided cells from 25% in the lowest iron condition (0.0002 mg/mL holotransferrin) to 61% (Fig. 6c). At 72 h, the fraction of cells which could undergo three or more divisions in low iron conditions was profoundly increased with aspartate supplementation (increased from 7% to 56% in the lowest iron condition; Fig. 6d). Together, these data indicate that aspartate alone can substantially overcome the cell cycle impairment induced by iron scarcity and promotes a more profound recovery of cell division compared to exogenous pyruvate or S*amhd1*-KO.

While clonal expansion is critical for effective CD8+ T cell effector responses, other aspects of CD8+ T cell biology provide important indicators of function. *Tfrc* mRNA expression was increased in low iron concentrations, indicative of cellular iron scarcity, irrespective of whether aspartate was supplemented or not (Supplementary Fig. 8a). However, aspartate supplementation did permit CD8+ T cells to maintain high TFR1/CD71 expression in the lowest iron concentrations when expression typically begins to drop (Fig. 6e). TFR1/CD71 levels may drop in the lowest iron conditions due to induction of tristetraprolin (TTP), a protein previously shown to suppress TFR1 expression under prolonged iron chelation[56] or may be due to decreased TFR1 protein synthesis under mTORC1 suppression (Fig. 2b, c). Aspartate may therefore provide benefit via promoting increased iron uptake capacity during iron starvation, although the mechanism is unclear. Independent of iron, aspartate supplementation also promoted CD8+ T cell expression of the activation marker, CD25, the cytolytic molecule, perforin and the cytokine, IFN-γ, indicating its availability within the cytosol supports multiple cellular processes beyond proliferation (Fig. 6f–h).

To understand if aspartate, in addition to altering cell proliferation and function, induces a shift in gene expression, we conducted an RNA-seq of CD8+ T cells in iron-replete and iron-deficient conditions, with or without aspartate (Supplementary Data 3). Samples were segregated largely based on iron concentrations rather than aspartate treatment (Supplementary Fig. 8b), suggesting that aspartate supplementation does not drive broad alterations in transcriptional profile. Aspartate supplementation induced relatively few transcriptional changes, with only 76 and 48 differentially expressed genes in high and low iron conditions, respectively. Notably, the transcriptional changes induced by aspartate in low iron conditions correlate with the changes mediated by aspartate in high iron conditions (Supplementary Fig. 8c), indicating aspartate has similar, albeit small, effects regardless of iron availability. Aspartate supplementation only marginally reduced *Cdkn1a* expression in low iron conditions (Supplementary Fig. 8d), indicating that the proliferative advantage conferred by aspartate is not entirely due to *Cdkn1a* suppression. GSEA revealed that the dominant aspartate effect is suppression of interferon (IFN) response pathways (Supplementary Fig. 8e) and was particularly driven by reduced IFN-stimulated gene expression. Similar effects were observed for aspartate treatment at high iron concentrations (Supplementary Fig. 8f). Notably, type-I IFN can have anti-proliferative effects that may be related to upregulation of *Cdkn1a*[57]. Thus, the rescue effect of aspartate on CD8+ T cell division may be partly facilitated if anti-proliferative type-I IFN signalling is counteracted.

Aspartate did not alter mROS generation (Supplementary Fig. 8g) but did increase mTORC1 activity measured via the expression of the downstream target, pS6 (Fig. 6i) and slightly suppressed the iron deficiency-mediated increase in the NAD+/NADH ratio (Supplementary Fig. 8h). Aspartate-treated CD8+ T cells also displayed increased glycolytic and total ATP production (Supplementary Fig. 8i–k) and increased the abundance of the purine and pyrimidine precursors, SAICAR and N-carbamoyl-aspartate, albeit not significantly, suggesting that aspartate partially rescues nucleotide production, whilst consumption likely also increases in these more highly proliferating cells (Fig. 6j, k). Finally, aspartate counteracted the accumulation of the repressive histone mark H3K27me3 by iron-deficient CD8+ T cells (Fig. 6l). Therefore, even in cells cultured in very low iron conditions, aspartate availability enables TCR-triggered CD8+ T cells to acquire an activated phenotype, reconfigure important aspects of their metabolism and proliferate.

## Discussion

Given the breadth and conserved nature of pathways involving iron-interacting proteins, it follows that iron deprivation would likely have broad implications on cell biochemistry, health and function, with potential knock-on effects at the tissue or systemic levels. Our work explains why iron depletion potently suppresses T cell responses in models of immunisation, infection and autoimmunity[5,26,30,58,59]. We also propose that the iron deficiency-associated perturbations described here may in part be responsible for cellular dysfunction in other cell types and tissues during iron deficiency.

Cells that undergo rapid proliferation are more likely to be impacted by iron deficiency than quiescent populations. This is because in iron-scarce environments, dividing cells will dilute out iron stores between daughter cells without exogenous iron pools to replenish the difference, and dividing cells have high biosynthetic demands. Meanwhile, quiescent cells can furnish iron-interacting proteins via internal recycling of iron cofactors. This principle is demonstrated by T cells homozygous for the TFR1[Y20H] mutation, which reduces iron uptake efficiency[11]. Quiescent naïve TFR1[Y20H] T cells are not noticeably impacted by the mutation but become extremely sensitive upon activation[11]. Moreover, some tissues, such as muscle, may be able to resist iron deficiency to some degree by upregulation of glycolysis[60]. However, the capacity to switch to a glycolytic programme may depend on having reduced demands for nucleotides and other macromolecular synthesis processes necessary for proliferation.

We demonstrate that iron deficiency in CD8+ T cells promotes an altered metabolic state. Iron deficiency caused relatively modest perturbation of the cellular transcriptome and proteome, at odds with the significant inhibition of proliferation. Iron deficiency acts most immediately to derail metabolism, with the most drastic effects on mitochondrial function. We observed decreased TCA cycle activity at the iron-dependent enzymes ACO2 and SDH, resulting in depletion of downstream metabolites. Despite suppressed TCA cycling, aspartate levels were augmented, but metabolites immediately downstream of aspartate metabolism, used for cytosolic nucleotide synthesis, were suppressed. We propose that aspartate accumulates within mitochondria due to iron-deficiency-associated suppressed activity of the proton-dependent mitochondrial aspartate carrier, SLC25A12/13. Remarkably, aspartate supplementation of culture media, assumed to access the cytosol directly, profoundly rescued iron-deficiency impaired proliferation and enhanced other functional aspects of activated T cells in both iron-dependent and -independent ways. Our results place disrupted aspartate handling front and centre in the metabolic derangements of cellular iron deficiency.

Iron deficiency induced profound alterations to the transcriptional and proteomic landscape with significant perturbations in metabolic gene expression, including suppressed mTORC1 pathway-associated genes and elevated OXPHOS and β-oxidation proteins. mTORC1 is essential for T cell metabolic reprogramming downstream of TCR stimulation for ATP and biosynthetic substrate generation required for activation[61]. Iron deficiency partially recapitulates the phenotype of mTORC1-inhibited CD8+ T cells, which feature reduced expression of ribosomal proteins[15]. However, in contrast to the decline in mitochondrial proteins observed with rapamycin[15], we instead saw increased mitochondrial proteins, indicating that iron deficiency's impact on metabolism is not simply due to suppression of mTORC1. We hypothesise that the accumulation of mitochondrial proteins in low iron conditions could be due to augmented generation of mitochondrial proteins to compensate for suppressed mitochondrial activity or impaired clearance by proteolysis or mitophagy.

Increased H3K27me3 in iron-deficient CD8+ T cells is suggestive of KDM6 dysfunction. The inability of CD8+ T cells to remodel their chromatin environment likely has implications for cellular differentiation. Moreover, suppressed levels of α-KG and/or iron may be capable of inhibiting iron and α-KG-dependent enzymes more generally, including other KDM enzymes and the ten-eleven translocation DNA demethylases, as has been seen for KDM3B, which can act as an

iron sensor influencing H3K9 methylation and mTORC1 activation[62], and the PHD proteins, which regulate HIF1α[28,63]. However, iron scarcity did not stabilise HIF1α or induce a proteomic response resembling that induced under hypoxic conditions. Different iron-dependent enzymes have been shown to have differential responses to iron inaccessibility, likely due to different iron binding affinities and expression levels[62]. For instance, KDM3B has a 40-fold lower affinity for iron than KDM3A ($KM_{[Fe]} = 100 \pm 30 \mu M$ and $KM_{[Fe]} = 2.6 \pm 0.8 \mu M$, respectively), providing rationale for why iron chelation results in accumulation of repressive H3K9me2 in a KDM3B, but not KDM3A, dependent manner[62]. Similarly, the PHD proteins have very high affinities for iron[64], potentially explaining our result that iron chelators, which potently suppress iron availability, induce a HIF1α response while iron deficiency caused by low extracellular transferrin-iron supply cannot. Thus, iron deprivation does not blanket inhibit all iron-dependent pathways, and iron chelation does not fully recapitulate physiological iron deficiency.

In 1925, Otto Warburg proposed that 'every living cell contains iron and that life without iron is impossible'[65], referring to the fact that cell culture media requires iron in order to support cellular growth[65]. Activated iron-deficient T cells do not progress through the proliferative cycle, but also do not apoptose, as has been observed by others[62]. Cellular iron deficiency mediated by depriving physiological sources of extracellular iron imparts a distinct and dysregulated metabolic signature. Our findings demonstrate that a lack of iron alters mitochondrial function, with knock-on effects influencing utilisation of cytosolic and nuclear metabolic pathways, ultimately affecting cell identity. Iron-deficient activated CD8+ T cells are 'stunned' but not moribund, and can be substantially restored by aspartate, partially overcoming Warburg's limitation. Aspartate is generally poorly taken up by cells, but our findings suggest that engineering cells, such as tumour-targeted chimeric antigen receptor T cells, to overexpress the plasma membrane aspartate transporter (SLC1A1/2/3) could provide resistance to iron depletion, for example, in niches such as the tumour microenvironment.

## Methods

### Ethics statement
This research complies with all relevant ethics regulations at the University of Oxford. Animal procedures were completed under the authority of the UK Home Office project and personal licences granted under the Animals (Scientific Procedures) Act (ASPA) of 1998.

### Mice
Mice were housed in individually ventilated cages and fed *ad libitum* A03 standard maintenance diet (SAFE diets, SAFE A03). The light/dark cycle was 12/12, and the temperature and humidity were maintained at 19–24 °C and 45–65%, respectively. OT-I CD45.1 mice on a C57BL6 background were acquired from Vincenzo Cerundolo, University of Oxford and Audrey Gerard, University of Oxford. *Samhd1*-KO mice on a C57BL6 background were acquired from Jan Rehwinkel[54]. C57BL6/J mice were purchased from Envigo. Mice of both sexes were used at 6–20 weeks of age. Sex was not considered in the experimental design. Mice were euthanised using a rising concentration of $CO_2$ followed by cervical dislocation.

### Cell culture media
To manipulate iron availability, iron-free medium was prepared with RPMI 1640 media (Gibco, 21875034), 1% penicillin/streptomycin (Sigma Aldrich, P0781-100ML), 1% glutamine (Sigma Aldrich, G7513-100ML) and 10% iron-free serum substitute (Pan Biotech, custom ordered as iron-free based on P04-95080). Iron-free media was supplemented with set concentrations of human holotransferrin (R&D Systems, 2914-HT-001G/Sigma Aldrich, T0665) ranging from 0.0002-0.625 mg/mL. Total transferrin was kept constant at 1.2 mg/mL by

adjusting human apotransferrin (unbound transferrin) concentrations (R&D Systems, 3188-AT-001G/Sigma Aldrich, T1147) accordingly. The highest concentration of 0.625 mg/mL holotransferrin (-15.6 μmol/L of iron) is reflective of the levels of iron physiologically found in human sera (14–32 μmol/L of iron[16]). Meanwhile, the low holotransferrin concentrations of 0.001 mg/mL are much lower than those found in plasma, but are hypothesised to be representative of highly iron-depleted environments such as those in the tumour microenvironment[48].

Aspartate (Scientific Laboratory supplies, CHE2306) was dissolved in iron-free media by slowly titrating in 1 M NaOH (Sigma Aldrich, 30620-1KG-M) until aspartate was completely solubilised and the pH was ~7.5. Asparagine and pyruvate were dissolved directly into iron-free media at the described concentrations.

### CD8+ T cell activation
Plates for CD8+ T cell activation were coated with 5 μg/mL α-CD3 (Biolegend, 100239) in phosphate-buffered saline (PBS) for 2–3 h at 37 °C. Lymph nodes and spleens (inguinal, axillary, brachial, cervical and mesenteric) from mice were sterilely collected in iron-free media and homogenised through 40 μm filters using EasySep buffer (Stem Cell Technologies, 20144) or an in-house alternative (PBS + 2% FBS + 1 mM EDTA (Invitrogen, AM9260G)) to produce single cell suspensions. CD8+ T cells were isolated from total homogenate using the EasySep Mouse CD8+ T cell (Stem Cell Technologies, 19853) isolation kit and EasyPlate EasySep magnets (Stem Cell Technologies, 18102) or Easy-Eights EasySep magnets (Stem Cell Technologies, 18103), according to the manufacturer's protocols. Cells were optionally stained with CTV (Invitrogen, C34557) for 8 min at 37 °C prior to culture. Cells were plated at $0.5 \times 10^6$ cells/mL in iron-free media supplemented with defined holotransferrin concentrations (see section "Cell culture media"). Cells were provided with 50 μM β-mercaptoethanol (BME, Gibco, 31350-010), 1 μg/mL α-CD28 (Biolegend, 102115) and 50 U/mL IL-2 (Biolegend, 575402). CD8+ T cells were cultured at 37 °C, 5% $CO_2$ for 24–72 h.

### RNA extraction
CD8+ T cells were collected from cell culture plates and washed twice with PBS; cell pellets were immediately snap frozen on dry ice. T cell pellets were resuspended in 350 μL of RLT+ buffer from the Qiagen RNeasy plus mini kit (Qiagen, 74136) and RNA was extracted according to the manufacturer's instructions. For RNA extracted for RNA-sequencing, samples were treated on-column with the Qiagen RNase-free DNase I set (Qiagen, 79254) according to the method described in Appendix D of the Qiagen RNeasy mini kit manual (Qiagen, 74104). RNA concentrations were determined using a Nanodrop one instrument (Thermofisher Scientific).

### cDNA synthesis and qPCR
cDNA was synthesised using the High-Capacity RNA-to-cDNA kit (Applied Biosystems, 4388950), and qPCR experiments were completed on a QuantStudio 7 flex real-time PCR system (Applied Biosystems, 4485701) using the Taqman gene expression master mix (Applied Biosystems, 4369016) and appropriate Taqman Gene Expression Assay (*B2m*, Mm00437762_m1; *Cdkn1a*, Mm04205640_g1; *Tfrc*, Mm00441941_m1). *B2m* was used as the endogenous control gene.

### RNA-sequencing
For RNA-sequencing, RNA quality was assessed using an Agilent high-sensitivity RNA ScreenTape (Agilent, 5067-5579) and corresponding sample buffer (Agilent, 5067-5580) on a 4200 TapeStation system (Agilent, G2991BA). Library preparation and bulk mRNA-sequencing to a depth of 30 million paired-end reads using the Illumina NovaSeq 6000 platform was conducted by Novogene.

Read alignment was conducted using RNA-star to the *Mus Musculus* genome (mm39), and features were annotated using Feature-Counts from the subread package. Differential gene expression analysis was conducted using EdgeR (R package) with the thresholds of $\log_2|\text{fold change}| > 1.5$ and an FDR < 0.05 applied. GSEA was completed using FGSEA (R package) and Hallmark pathways with a significance threshold set as FDR < 0.05.

## ChIP-mentation

CD8+ T cells were activated for 48 h, after which they were collected, washed twice in PBS and fixed in 1% paraformaldehyde (PFA, Thermo Scientific, 28906) for 10 min at room temperature with rotation. Cells were washed twice at $16,000 \times g$ for 30 s, and dry pellets were snap frozen on dry ice.

Antibody binding buffer was prepared: 0.5% BSA and 1/200 protease inhibitor cocktail (PIC) in PBS. 10 μL of protein A dynabeads (ThermoFisher, 1001D) per sample were washed twice with antibody binding buffer on a magnet. Beads were blocked with 1 μL of antibody (α-H3K27ac, Diagenode, C15410196; α-H3K4me3, Diagenode, C15410003; α-H3K27me3, Sigma Aldrich, 07-449) in antibody binding buffer for 3–4 h at 4 °C with rotation. Just prior to adding the chromatin, the beads were washed with antibody binding buffer.

Cell pellets were resuspended in 120 μL of lysis buffer (50 mM Tris pH 8, 10 mM EDTA, 0.5% SDS) with 1/200 PIC and transferred into Covaris tubes (Covaris, 520045). Samples were sonicated for 180 s using the settings aiming for 200 bp fragments. Samples were diluted with lysis buffer with 1/200 PIC to a 900 μL total volume, and 100 μL of 10% Triton X-100 was added to neutralise. Samples were incubated for 10 min with rotation. 10 μL of prewashed protein A dynabeads were added and incubated for 30 min at 4 °C. Samples were placed on a magnet, and 250 μL of sample supernatant was added to antibody-bound beads. Chromatin/antibody/bead mixtures were incubated overnight at 4 °C with rotation. Samples were washed three times with 150 μL RIPA wash buffer (50 mM HEPES pH 7.6, 500 mM LiCl, 1 mM EDTA, 1% NP-40, 0.7% NaDeoxycholate), once with 150 μL TE buffer and once with 150 μL 10 mM Tris pH 8, each time for 5 min. Beads were resuspended in 29 μL pre-warmed tagmentation buffer with 1 μL Tn5 transposase (Illumina, 20034197). Samples were incubated for 5 min with vigorous mixing at 1100 rpm at 37 °C in a thermomixer. 150 μL of RIPA buffer was added immediately to degrade the Tn5. Beads were washed once with 150 μL of RIPA buffer and then resuspended with 22.5 μL of ddH₂O.

25 μL of 2X NEBNext Ultra II Q5 MasterMix (New England Biosciences, M0544S), 1.25 μL 5 μM universal adaptor primer and 1.25 μL 5 μM index adaptor primer was added to each sample. Samples were transferred to a thermocycler and the following programme was run: 72 °C for 5 min, then 95 °C for 5 min followed by 12 cycles of 98 °C for 10 min, 63 °C for 30 s and 72 °C for 3 min. Following the 12 cycles, samples were incubated at 72 °C for 5 min and then held at 12 °C. Supernatants were removed and 50 μL of Ampure XP beads (Beckman Coulter, A63880) were added and pipetted 10 times and then incubated at room temperature for 2 min. Beads were washed once with 150 μL 80% ethanol, incubating for 2 min at room temperature. The ethanol was removed, and the beads were allowed to air dry for 3–5 min. 11 μL of ddH₂O was added to the beads to elute the DNA. The supernatant was collected.

DNA quality was assessed using an Agilent D100 high-sensitivity screentape (Agilent, 5067-5584) and the High-sensitivity D1000 reagents (Agilent, 5067-5585) on a 2200 TapeStation system (Agilent, G2964AA) and using a Qubit fluorometer. DNA sequencing to a depth of 10 million paired-end reads using the Illumina NovaSeq 6000 platform was conducted by Novogene.

Data was analysed using the SeqNado analysis pipeline (https://github.com/alsmith151/SeqNado). Peak enrichment for H3K27ac and H3K4me3 was conducted for the promoters of genes previously identified as being differentially regulated at the RNA level by the RNA-seq.

## Protein-mass spectrometry

CD8+ T cells were activated for 48 h, harvested and washed twice in PBS. Cells were fixed in 2% PFA (Thermo Scientific, 28906) for 30 min, washed and resuspended in PBS. Cells that were alive at the time of fixation were sorted by flow cytometry using forward and side scatter into PBS. Cells were pelleted and snap frozen on dry ice. $N = 4$ for each condition for the protein-MS experiment. Pelleted cells were lysed using the 2-step trypsin lysis protocol described by Kelly et al.[66]. Cell pellets were resuspended in 200 μL TEAB digest buffer (0.1 M TEAB, 1 mM MgCl2, 1:80 benzonase, pH 8) and incubated for 20 min at 37 °C in a thermomixer. The amount of trypsin for a 1:20 trypsin to protein (w/w) ratio was calculated, and 50% of the required trypsin was added. Samples were incubated overnight at 37 °C in a thermomixer. The remaining trypsin was added, and samples were incubated for 60 min at 37 °C in a thermomixer. The samples were acidified to a final concentration of 1% trifluoroacetic acid (TFA) and subjected to a C18 stage-tip desalting with the following buffers: condition (100% acetonitrile), wash (0.1% TFA), elute (66.6% acetonitrile, 0.1% TFA), ion exchange (2:1 ratio of 100% acetonitrile to 0.1% TFA). Samples were dried using a SpeedVac at 65 °C.

Protein-MS analysis was performed as described previously[67]. Briefly, peptides were analysed on a Q-Exactive-HF-X (Thermo Scientific) mass spectrometer coupled with a Dionex Ultimate 3000 RS (Thermo Scientific). LC buffers were the following: buffer A (0.1% formic acid in Milli-Q water (v/v)) and buffer B (80% acetonitrile and 0.1% formic acid in Milli-Q water (v/v)). 2 μg of each sample were loaded at 15 μL/min onto a trap column (100 μm × 2 cm, PepMap nanoViper C18 column, 5 μm, 100 Å, Thermo Scientific) equilibrated in 0.1% TFA. The trap column was washed for 3 min at the same flow rate with 0.1% TFA, then switched in line with a Thermo Scientific, resolving C18 column (75 μm × 50 cm, PepMap RSLC C18 column, 2 μm, 100 Å). The peptides were eluted from the column at a constant flow rate of 300 nL/min with a linear gradient from 3% buffer B to 6% buffer B in 5 min, then from 6% buffer B to 35% buffer B in 115 min, and finally to 80% buffer B within 7 min. The column was then washed with 80% buffer B for 4 min and re-equilibrated in 3% buffer B for 15 min. Two blanks were run between each sample to reduce carry-over. The column was always kept at a constant temperature of 50 °C.

The data were acquired using an easy spray source operated in positive mode with spray voltage at 1.9 kV, the capillary temperature at 250 °C, and the funnel RF at 60 °C. The MS was operated in data-independent acquisition (DIA) mode as reported earlier with some modifications[68]. A scan cycle comprised a full MS scan ($m/z$ range from 350 to 1650, with a maximum ion injection time of 20 ms, a resolution of 120,000, and an automatic gain control (AGC) value of $5 \times 10^6$). MS survey scan was followed by MS/MS DIA scan events using the following parameters: default charge state of 3, resolution 30,000, maximum ion injection time 55 ms, AGC $3 \times 10^6$, stepped normalised collision energy 25.5, 27 and 30, fixed first mass 200 $m/z$. Data for both MS and MS/MS scans were acquired in profile mode. Mass accuracy was checked before the start of sample analysis.

Quantification of reporter ions was completed using Spectronaut (Biognosys; Spectronaut 14.10.201222.47784) in library-free (direct-DIA) mode. Minimum peptide length was set to 7, and maximum peptide length was set to 52, with a maximum of 2 missed cleavages. Trypsin was specified as the digestive enzyme used. The FDR at the precursor ion level and protein level was set at 1% (protein and precursor Q value cutoff). The max number of variable modifications was set to 5, with protein N-terminal acetylation and glutamine and asparagine deamidation, and methionine oxidation set as variable modifications. Carbamidomethylation of cysteine residues was selected as a fixed modification. Data filtering and protein copy number

quantification were performed in the Perseus software package, version 1.6.6.0. Copy numbers were calculated using the proteomic ruler[69]. This method sets the summed peptide intensities of the histones to the number of histones in a typical diploid cell. The ratio between the histone peptide intensity and summed peptide intensities of all other identified proteins is then used to estimate the protein copy number per cell for all the identified proteins. A $\log_2$|fold change| > 0.585 (equivalent to |fold change| > 1.5) was used (as used in Howden et al.[15]). We also applied the typical $p$ value threshold of <0.05 when comparing the high (0.625 mg/mL holotransferrin) and low (0.001 mg/mL holotransferrin) iron concentrations via a $t$-test as well as an additional threshold of $p$ values < 0.05 when a one-way ANOVA was conducted across all conditions. GSEA was conducted using FGSEA (R package) and the Hallmark pathways. For analysing mitochondrial proteins, proteins were filtered by inclusion in the Mito-Carta3.0 gene set[31].

## $^{13}$C6-glucose and $^{13}$C5-glutamine tracing

For heavy isotope tracing experiments, CD8+ T cells were isolated and activated as described above. Iron-free media with varying holotransferrin concentrations were prepared using phenol red-free RPMI 1640 (Gibco, 11835030) as phenol red can interfere with metabolite-ms. At 24 h prior to cell collection (24 h post-activation), cells were collected, washed and replated in media containing the heavy isotope of interest (13C6-glucose (Cambridge Isotope Laboratories, Inc., CLM-1396-1) and 13C5-glutamine (CK Isotopes, CNLM-1275)) on plates coated with α-CD3. Iron-free tracing media was prepared using SILAC RPMI 1640 flex media (Gibco, A24945201), which lacks glucose, phenol red, glutamine, arginine and lysine. Arginine (MP Biomedicals, 194626) and lysine hydrochloride (MP Biomedicals, 194697) were supplemented at standard RPMI 1640 concentrations of 1.2 mM and 0.2 mM, respectively. For 13C6-glucose tracing, glutamine (Sigma Aldrich, G7513-100ML) was added at 2 mM and 13C6-glucose at 11.1 mM. For 13C5-glutamine tracing, 13C5-glutamine was added at 2 mM and glucose (Sigma Aldrich, 158968-100G) at 11.1 mM. Holotransferrin and apotransferrin, as well as activation reagents (BME, α-CD28 and IL-2) were added to the iron-free tracing media as described under 'CD8+ T cell activation' at standard concentrations. $N = 4$ for each condition for the carbon tracing experiments. Metabolites were measured by Gas chromatography-mass spectrometry as described below.

## Gas chromatography-mass spectrometry (GC/MS) for metabolites

Forty-eight-hour activated CD8+ T cells were collected, counted and $2–4 \times 10^6$ cells (experiment dependent, however, the same number of cells were used for all samples within an experiment) were washed twice in ice-cold 0.9% saline made with NaCl (Sigma Aldrich, 31434-500G-M) and ultrapure HPLC grade water (Alfa Aesar, 22934). Cells were pelleted and snap frozen on dry ice. Cells were extracted in 1:1:1 pre-chilled methanol, HPLC grade water (containing 1.75 μg/mL D6-glutaric acid) and chloroform. The extracts were shaken at 1400 rpm for 15 min at 4 °C and centrifuged at $12,000 \times g$ for 15 min at 4 °C. The upper aqueous phase was collected and evaporated under vacuum. Metabolite derivatization was performed using an Agilent autosampler. Dried polar metabolites were dissolved in 15 μL of 2% methoxyamine hydrochloride in pyridine (Thermo Fisher Scientific, 25104) at 55 °C, followed by an equal volume of N-tert-Butyldimethylsilyl-N-methyltrifluoroacetamide with 1% tertbutyldimethylchlorosilane after 60 min, and incubation for a further 90 min at 55 °C. GC-MS analysis was performed using an Agilent 6890 GC equipped with a 30 m DB-35 MS capillary column. The GC was connected to an Agilent 5975C MS operating under electron impact ionisation at 70 eV. The MS source was held at 230 °C and the quadrupole at 150 °C. The detector was operated in scan mode, and

1 μL of derivatised sample was injected in splitless mode. Helium was used as a carrier gas at a flow rate of 1 mL/min. The GC oven temperature was held at 80 °C for 6 min and increased to 325 °C at a rate of 10 °C/min for 4 min. The run time for each sample was 59 min. For the determination of the mass isotopomer distributions, spectra were corrected for natural isotope abundance. Data processing was performed using MATLAB.

## Liquid chromatography-mass spectrometry (LC/MS) for AICAR detection

CD8+ T cells were collected after 48 h of culture, and $3 \times 10^6$ cells were washed twice with PBS. $N = 4$ for each condition for the AICAR detection experiment. Dry pellets were snap frozen on dry ice. Cell pellets were resuspended in 0.01% formic acid (Sigma Aldrich, 1003445799) containing 0.8 μM 15N2-dT (Cambridge Isotope Labs, NLM-3901-PK) and were incubated at 37 °C for 30 min. Samples were centrifuged at maximum speed for 10 min, and the lysates were collected and added to 3 kDa Amicon Ultra 0.5 centrifugal filter units (Millipore, UFC500324) and centrifuged for 20 min at $16,000 \times g$. The filtrates were transferred to a mass spectrometry vial for analysis of compounds in nucleotide synthesis.

Samples were analysed on a TSQ Altis Triple Quadrupole Mass Spectrometer in selected reaction monitoring mode, interfaced to an UltiMate 3000 uHPLC. The uHPLC was fitted with a nanoEase M/Z Symmetry C18 Trap Column, 100 Å, 5 μm, 180 μm × 20 mm (Waters) at RT and a Luna Omega 3 μm PS C18 column (150 × 0.3 mm) connected to an EASY-Spray™ source with an EASY-Spray Cap flow emitter (15 μM). Ten microlitres of sample were injected per run via a 10 μL sample loop. Buffers used were from Romil and of Ultra LC standard. Buffer A: $H_2O$ (0.1% acetic acid), buffer B: MeCN (0.1% acetic acid). The gradient was 0–2.8 min–1% B, 22 min–30% B, 23.5 min–99% B. This was followed by 2 wash pulses (1–99% B) and equilibration to 1% B (45 min total run time). The trap column was held at a constant 1% B, and switching from the trap to the main column occurred at 1 min and back at 40 min. Mass spectrometry conditions were as follows: source voltage of 2400 V in positive ionisation mode; ion transfer tube temperature 275 °C, CID gas pressure 1.5 mTorr, scan widths for Q1 and Q3 at 0.7 $m/z$, a chromatographic filter was used with a peak width of 6 s, dwell time 20 ms. Collision energy voltage, and RF voltage were optimised using the Thermo Tune software for the compounds assayed using authentic compounds dissolved in Buffer A. Peaks retention times were confirmed in samples by co-injection with the relevant standard.

Data was analysed using the FreeStyle 1.6 software and the Genesis peak detection algorithm with its default settings. Data was analysed by ratioing the target peak area with the $^{15}N_2$-dT peak area.

## Liquid chromatography-mass spectrometry (LC/MS) for nucleotides and nucleotide precursors

CD8+ T cells were collected at 48 h post-activation, and $3 \times 10^6$ cells were washed twice with ice-cold 0.9% saline made with NaCl (Sigma Aldrich, 31434-500G-M) and ultrapure HPLC grade water (Alfa Aesar, 22934). Cells were pelleted and snap frozen on dry ice. $N = 4$ for each condition for LC-MS experiments. Metabolites were extracted with 100 μL of ice-cold extraction buffer (40% methanol (Biosolve, BIO-13687802), 40% acetonitrile (Biosolve, BIO-01204102), 20% water (Biosolve, BIO-23214102-1), 15 μM glutaric acid internal standard and 0.5% formic acid (Biosolve, BIO-069141A8)) for 5 min on ice. 8.8 μL of 15% ammonium bicarbonate (Supelco, 5.33005) solution was added to neutralise, and samples were left on dry ice for a further 15 min. Samples were thawed on ice and then centrifuged at 20 G at 4 °C for 5 min. Supernatants were collected for LC-MS analysis.

LC-MS analysis was performed on an Agilent LC-MS QToF 6546 using a Waters Premier BEH Z-HILIC VanGuard Fit column (1.7 μm, 2.1 × 150 mm) with the following mobile phases:

Mobile phase A: 20 mM Ammonium hydrogen carbonate (LiChropur, Supelco) in LC-MS grade water (Ultra CHROMASOLV, Honeywell Riedel-de Haën) with 0.1% ammonium hydroxide (Alfa Aesar, Thermo Scientific) and 5 μM InfinityLab deactivator additive (Agilent Technologies).

Mobile phase B: 90% UHPLC grade acetonitrile (BioSolv, Greyhound Chemicals), 10% LC-MS grade water (Ultra CHROMASOLV, Honeywell Riedel-de Haën) with 5 μM InfinityLab deactivator additive (Agilent Technologies).

Five microlitre of sample was injected and the chromatographic separation was achieved with a gradient run with a constant flow rate of 0.2 mL/min and the following programme: $T = 0$ min, 10% A, 90% B; $T = 2$ min, 10% A, 90% B; $T = 18$ min, 35% A, 65% B; $T = 22$ min, 65% A, 35% B; $T = 22.1$ min, 85% A, 15% B; $T = 25$ min, 85% A, 15% B; $T = 25.1$ min, 10% A, 90% B; $T = 30$ min, 10% A, 90% B.

Full scan data was acquired between $m/z$ 50–1050 at 1 Hz whilst using online mass correction. Analyte ionisation was achieved via ESI (negative polarity) with the following parameters: VCap: 2000 V, Nozzle Voltage: 500 V, gas temperature: 225 °C, drying gas: 8 L/min, nebuliser: 30 psi, sheath gas temp: 300 °C, sheath gas flow 12 L/min. Data extraction was performed using Agilent Profinder 10.0. Data were normalised to the internal standard (D6-Glutaric acid) peak area.

## NAD+/NADH ratio measurements

NAD+/NADH ratio measurements were made using the NAD/NADH-Glo assay (Promega, G9071) according to the manufacturer's instructions with modifications for measuring NAD+ and NADH individually (rather than pooled). Cultured CD8+ T cells were collected, washed in PBS and counted. $1 \times 10^5$ cells in 50 μL of PBS were lysed in 50 μL of 0.2 M NaOH (Sigma Aldrich, 30620-1KG-M) with 1% dodecyltrimethylammonium bromide (DTAB, Alfa Aesar, A10761) per condition in technical duplicates. Each sample was split (50 μL) into a second tube for matched NAD+ and NADH measurements.

For measurement of NAD+, NADH was first degraded by adding 25 μL of 0.4 M HCl (Fisher Scientific, H/1111/PB17) to the first tube, heated to 60 °C for 15 min followed by incubation at room temperature for 10 min. Twenty-five microlitres 0.5 M TRIZMA base (Sigma Aldrich, T1503-1KG) were added to quench the HCl.

To measure NADH, NAD+ was degraded by heating the second tube to 60 °C for 15 min followed by incubation at room temperature for 10 min. Fifty microlitres of 0.2 M HCl (Fisher Scientific, H/1111/PB17) and 0.25 M TRIZMA base (Sigma Aldrich, T1503-1KG) solution were added to quench the NaOH.

The NAD/NADH-Glo detection reagent was prepared as instructed. Fifty microlitres of each sample (either purified for NAD+ or NADH) and 50 μL of NAD/NADH-Glo detection reagent were added to a white 96-well plate and mixed gently. The plate was incubated at room temperature and luminescence was measured using a Promega GloMax multi-detection luminometer at 30, 45 and 60 min.

## Extracellular flux analysis assay

Extracellular flux analysis (Seahorse) was conducted using the Seahorse XF Real-Time ATP rate assay kit (Agilent, 103592-100). The day prior to the assay, the Agilent Seahorse XF96 analyser was turned on, and a sensor cartridge (Agilent, 103792-100) was hydrated in sterile dH2O at 37 °C in a non-CO2 incubator overnight. The water was replaced by Seahorse XF calibrant (Agilent, 103059-000) on the day of the assay, and the cartridge was incubated for 1 h prior to loading the drug ports. The day of the assay, ATP assay media was prepared using 1 mM pyruvate (Agilent, 103578-100), 2 mM glutamine (Agilent, 103579-100), 10 mM glucose (Agilent, 103577-100) in Seahorse XF RPMI pH 7.4 (Agilent, 103576-100) and warmed to 37 °C. Cells were collected, washed and $1 \times 10^5$ cells in 50 μL of ATP assay media were plated per well on a Poly-D-lysine (Gibco, A3890401) coated XF96 microplate (Agilent, 101085-004). Plates were spun down,

supernatant removed and cells incubated at 37 °C for 30 min in a non-CO2 incubator, followed by the addition of 130 μL of ATP assay media and further incubation for 25–30 min. Oligomycin and rotenone/antimycin A were used at a final concentration of 1.5 μM and 0.5 μM, respectively. Twenty microlitres of oligomycin (15 μM) and 22 μL of rotenone/antimycin A (5 μM) were loaded into the sensor cartridge in ports A and B, respectively. The ATP rate assay was run on a Seahorse XF96 analyser with 3 measurements taken for each step (baseline, oligomycin, rotenone/antimycin A) using the following injection settings: 3 min mix, 0 min wait, 3 min measure. Data were analysed using the Agilent Seahorse Wave desktop software (version 2.6) and the Seahorse XF real-time ATP rate assay report generator.

## Flow cytometry

Cells were transferred from cell culture plates to 96-well round-bottom plates and washed with PBS.

In intracellular cytokine staining experiments, cells were first stimulated with cell activation cocktail (1:2500) (Biolegend, 423301), BFA (5 mg/mL) (Biolegend, 420601) and monensin (2 mM) (Biolegend, 420701) in iron-free media at 37 °C/5% CO2 for 5 h prior to staining.

Cells were stained with 20 μL of surface antibody cocktail with Zombie NIR fixable viability kit (1:400-1000, Biolegend, 423105) in PBS for 20 min on ice. Cells were fixed with 2% paraformaldehyde (Pierce, 28906) diluted in PBS or commercial fixation buffer (Biolegend, 420801) for 20 min on ice. For nuclear staining of markers such as H3K27me3, cells were fixed in Foxp3 transcription factor fixation buffer (eBioscience, 00-5523-00) for 1 h on ice.

For intracellular staining, cells were permeabilised in perm/wash buffer (Biolegend, 421002) for 20 min. Intracellular targets were stained with 20 μL of intracellular stain prepared in perm/wash buffer for 20 min to overnight. Cells were resuspended in PBS, and data were acquired using an Attune NxT flow cytometer (Thermofisher Scientific). Data was analysed using FlowJo (BD Biosciences). A representative gating scheme can be found in Supplementary Fig. 9.

Detection of mROS was conducted using MitoSOX dye (Invitrogen, M36008). Cells were resuspended in 200 μL of MitoSOX dye diluted to 5 μM in phenol red iron-free media and incubated at 37 °C/5% CO2 for 15 min before surface staining. The acquisition was conducted on live cells.

Cells were stained with MTG (M7514, Thermofisher Scientific) and Mitotracker Deep Red (MTDR; M22426, Thermofisher Scientific) at 100 nM in iron-free media for 30 min at 37 °C/5% CO2 prior to staining. Cells were acquired live. The ratio of MTDR to MTG was calculated as a metric of mitochondrial membrane potential relative to mitochondrial mass.

For flow staining for the metabolic regulators, pS6 and HIF1α, cells were washed once with a solution of 50% RPMI 1640, 50% HBSS (Gibco, 14025-092) and 0.5% BSA (PAN biotech, PO6-139310) before fixation in 1% PFA (Pierce, 28906) for 20 min. Cells were washed with PBS and optionally surface-stained for fixation of stable epitopes. Cells were permeabilised using 90% ice-cold methanol (Sigma Aldrich, 34860-1L-R) for 20 min at −20 °C prior to permeabilization and acquisition.

Antibodies used: CD25-APC (Biolegend, PC61, 102011, 1:200), CD25-PE-Cy7 (Biolegend, PC61, 102015, 1:200), CD44-BV605 (Biolegend, IM7, 103047), CD71-FITC (Biolegend, RI7217, 113805, 1:400), CD71-APC (Biolegend, RI7217, 113819, 1:200), CD71-PE-Cy7 (Biolegend, RI7217, 113811, 1:200), CD71-PE (Biolegend, RI7217, 113807, 1:200), CD8a-FITC (Biolegend, 53-6.7, 100705, 1:400), CD8a-PE (Biolegend, 53-6.7, 100707, 1:200), CD8a-PerCP-Cy5.5 (Biolegend, 53-6.7, 100733, 1:200), CD98-APC (Biolegend, RL388, 128211, 1:200), CD107a-FITC (Biolegend, 1D4B, 121606, 1:400), GZMB-FITC (Biolegend, GB11, 515403, 1:100), H3K27me3-AF488 (Cell Signalling, C36B11, 5499S, 1:50), IFNg-Pe-Cy7 (Biolegend, XMG1.2, 505825, 1:200), IL-2-PE (Biolegend,

JES6-5H4, 503807, 1:200), Perforin-PE (Biolegend, S16009B, 154405, 1:200), pS6-AF647 (Cell Signalling, D57.2.2E, 14733S, 1:50), TNFa-FITC (Biolegend, MP6-XT22, 506603, 1:400).

## Cell counting by flow cytometry

Cells were transferred into 96-well round-bottom TC plates and washed once with 2% FBS in PBS (FACS buffer). Cells were resuspended in FACS buffer containing 0.5 µg/mL 7-AAD (Biolegend, 420403) and incubated on ice for 10 min and then acquired directly on the Attune NxT flow cytometer (Thermofisher Scientific). Fifty microlitres of cells were acquired per well to enable accurate calculation of cell concentration, and wells were resuspended using a multichannel pipette between each column on the 96-well plate to account for cell settling effects.

## Data analysis

Unless explicitly specified, data analysis was completed using Excel (Microsoft), Prism (GraphPad), FlowJo (BD Biosciences), or published R packages. Statistics are specified in the figure legends.

## Reporting summary

Further information on research design is available in the Nature Portfolio Reporting Summary linked to this article.

## Data availability

The RNA-seq and ChIPmentation datasets in this study have been deposited in the Gene Expression Omnibus (GEO) under the accession codes GSE251962, GSE251963, GSE251964 and GSE292414. The protein-MS dataset in this study has been deposited in the ProteomeXchange Consortium via the PRIDE[70] partner repository under the accession code PXD047814. The metabolomics dataset for measuring AICAR has been deposited in the MetaboLights repository here: https://metabolights-labs.org/u/christopher_lee_millington/h/megan-teh-raw-lc-ms-data. The remaining metabolomics datasets have been deposited to the MassIVE repository under the accession codes MSV000097525 (https://massive.ucsd.edu/ProteoSAFe/dataset.jsp?task=ec2369d9e92b40acabce8f292df2cc2c), MSV000097526 (https://massive.ucsd.edu/ProteoSAFe/dataset.jsp?task=db3cd5a35e034026ad6d73b793eea3ee), MSV000097527 (https://massive.ucsd.edu/ProteoSAFe/dataset.jsp?task=dbd7ebd8ec454ea1ad0312b8b90ba981) and MSV000097528 (https://massive.ucsd.edu/ProteoSAFe/dataset.jsp?task=bf2885289e684df2bcfb3e04854a6cc0). The reused proteomics dataset PXD012058 was from Proteome Exchange with the primary accession number PXD012058. All other data are available in the article and its Supplementary files or from the corresponding author upon request. Source data are provided with this paper.

## Code availability

The SeqNado pipeline used for analysing the ChIPmentation experiments is available at https://github.com/alsmith151/SeqNado.

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

## Acknowledgements

Thank you to the staff of the Department of Biomedical Services, University of Oxford for animal husbandry and welfare, the members of the Drakesmith lab, Michael Murphy (MRC Mitochondrial Biology Unit, University of Cambridge), Sumana Sharma (MRC Translational Immune Discovery Unit, University of Oxford), Katja Simon (The Kennedy Institute of Rheumatology, University of Oxford) and Ana-Victoria Lechuga-Vieco (The Kennedy Institute of Rheumatology, University of Oxford) for helpful discussions, Natasha White and Giulia Pironaci (both MRC Translational Immune Discovery Unit, University of Oxford) for assistance with OT-I colony management, Dana Costigan (MRC Translational Immune Discovery Unit, University of Oxford) for practical help with pilot experiments, Andrew Howden and the FingerPrints Proteomic Facility for protein-MS analysis (University of Dundee) and the support and resources of the Birmingham Metabolic Tracer Analysis Core (University

of Birmingham). M.R.T., J.N.F., A.E.P., H.M., A.E.A. and H.D. were supported by a U.K. Medical Research Council, MRC Human Immunology Unit grant awarded to H.D. (MCU_12010/10). J.N.F. is supported by a Cancer Research Institute fellowship (award CRI4186) and the Center for Experimental Immuno-Oncology of Memorial Sloan Kettering Cancer Center. M.R.T. was funded by the Clarendon Fund and the Corpus Christi College A. E. Haigh graduate scholarship. T.A.M. and A.S. were supported by a Medical Research Council (MRC, UK) Molecular Haematology Unit grant awarded to T.A.M. (MC_UU_00016/6 and MC_UU_00029/6). L.V.S. was supported by a Wellcome Trust Principal Research Fellowship, awarded to Doreen Cantrell (205023/Z/16/Z). N.G. and S.D. are supported by a Blood Cancer UK Project grant (21007) and an MRC New Investigator Research Grant (MR/V011588/1). J.R., B.M. and D.A.T. are supported by a Cancer Research UK Programme grant (C42109/A24757).

## Author contributions

Conceptualisation: M.R.T., H.D., S.D., J.N.F., A.E.A., L.V.S., B.K., A.L.S., D.A.T, A.E.P., S.J.D., T.A.M., J.R.; methodology: M.R.T., N.G., L.V.S, A.L.S.; formal analysis: M.R.T.; investigation: M.R.T, N.G., J.N.F., L.V.S., A.L.S., C.L.M., B.K., J.R., B.P.M., H.M., V.S.; resources: J.R.; Writing, M.R.T, H.D., S.D., A.E.A.; supervision: H.D., A.E.A., S.D.; funding acquisition: H.D., S.D.

## Competing interests

T.A.M. is a paid consultant for and shareholder in Dark Blue Therapeutics Ltd. D.A.T. undertakes paid consultancy work for Sitryx Ltd. The remaining authors declare no competing interests.

## Additional information

[1]MRC Translational Immune Discovery Unit, Radcliffe Department of Medicine, MRC Weatherall Institute of Molecular Medicine, University of Oxford, Oxford, UK. [2]Institute of Immunology and Immunotherapy, University of Birmingham, Birmingham, UK. [3]Division of Cell Signalling and Immunology, University of Dundee, Dundee, UK. [4]MRC Molecular Haematology Unit, Radcliffe Department of Medicine, MRC Weatherall Institute of Molecular Medicine, University of Oxford, Oxford, UK. [5]Nuffield Department of Clinical Medicine, NDM Centre for Global Health Research, University of Oxford, Oxford, UK. [6]School of Medical Sciences, College of Medicine and Health, Institute of Metabolism and Systems Research, College of Medical and Dental Sciences, University of Birmingham, Birmingham, UK. [7]Mahidol-Oxford Tropical Medicine Research Unit, Mahidol University, Bangkok, Thailand. [8]Present address: CeMM—Research Center for Molecular Medicine of the Austrian Academy of Sciences, Vienna, Austria. [9]These authors contributed equally: Sarah Dimeloe, Hal Drakesmith. ✉e-mail: mteh@cemm.oeaw.ac.at; alexander.drakesmith@ndm.ox.ac.uk

