## [Transparent Peer Review file · Nature Communications]

Iron deficiency causes aspartate-sensitive dysfunction in CD8⁺ T cells

Corresponding Author: Professor Hal Drakesmith

Version 0:

Reviewer comments:

Reviewer #1

(Remarks to the Author)

This study employs an in vitro model to investigate the influence of iron on CD8 T cell activation. The researcher's findings suggest that iron deficiency diminishes cell proliferation, modifies histone methylation status, and disrupts mitochondrial membrane integrity. Moreover, there is inadequate utilization of glucose and glutamine in the TCA cycle as revealed by the metabolic profile. The author impressively adopts a multi-omics approach to characterize epigenetic regulation, gene expression, protein abundance, and metabolite levels. Nevertheless, the manuscript lacks coherent organization and focus. Additionally, since this concept has not been tested using an in vivo or genetic model it reduces enthusiasm for this study. Furthermore, the mechanism underlying reduced cell division due to iron deficiency remains unclear. Several concerns arise:

- 1) The researcher did not analyze any T-cell activation (CD44, CD69, and CD25) or functional markers until Figure 6. This makes it challenging to conclude the impact on T cell function. According to Figure 6, the absence of iron only decreases cell proliferation without affecting perforin and IFN γ levels. In the low holotransferrin group, less than 40% of cells were activated and divided; however, there was no change in overall perforin and IFN γ expression. This aspect is perplexing.
- 2) The False Discovery Rate for Figure 1F should be provided by the author.
- 3) An explanation is needed from the Author about why p53 regulated by iron only affects cell division without impacting survival measures.
- 4) The inclusion of the "iron-sufficient" group is required in Figure 3.
- 5) Figure 4 illustrates that the general amount of H3K27me3 is comparable in IL-7 naive and iron-deficient conditions. To demonstrate the overlap of the H3K27me3 site between these two states, conducting CHIPseq is essential for the author.
- 6.) Supplying α -ketoglutarate does not rescue phenotype needs clarification.

Reviewer #2

(Remarks to the Author)

This study investigates mechanisms by which iron deficiency impairs CD8⁺ T cell proliferation. The authors developed an elegant cell culture model for OT-I CD8⁺ T cells in the presence of variable concentrations of holo-transferrin. Iron limitation hampered cell proliferation without impacting cell viability. It also altered gene expression and disrupted mitochondrial functions. Iron-deficient CD8⁺ T cells had elevated levels of aspartate, a key mitochondria-derived metabolite. However, this aspartate wasn't utilized for nucleotide synthesis, likely due to trapping within depolarized mitochondria. Supplementation with exogenous aspartate, which directly accesses the cytosol, rescued clonal expansion of iron-deficient CD8⁺ T cells. These data validate the importance of electron transport chain (ETC)-linked aspartate synthesis in cell proliferation, as proposed in references 35-36 and, furthermore, provide an important link to iron metabolism. The findings are novel, and the experiments are rigorously designed and controlled. The conclusions are largely supported by the data, but there are issues requiring attention.

- 1) The induction of ETC and β -oxidation proteins, as well as of heme and Fe-S biosynthetic enzymes in iron-deficient cells is counterintuitive. Why would the cells waste so much energy when a critical co-factor is limiting? While the data show that

limited iron supply triggers cellular responses to iron deficiency (Tfrc induction), there is no evidence for mitochondrial iron deficiency. Cytosolic and mitochondrial iron levels should be quantified.

- 2) "RNA expression changes observed during iron deficiency are largely due to transcriptional differences rather than alterations in RNA stability". Does this apply to ETC and β -oxidation proteins, and heme and Fe-S biosynthetic enzymes?
- 3) Is the induction of Tfrc under iron-restricted conditions likewise transcriptional? Did the proteomics analysis identify increased IRP2 levels in iron-deficient cells? This is a good marker of post-transcriptional responses to iron deficiency. Why is Tfrc not included among the list of upregulated genes/proteins in supplementary Fig. 1h?
- 4) The cell cycle arrest observed in iron-deficient cells is attributed to CDKN1A induction. However, aspartate supplementation that rescued cell proliferation "only marginally reduced Cdkn1a expression in low iron conditions". Iron deficiency is also known to affect other cell cycle check points; for instance, p21Cip1/Waf1 (PMID 17429006). Could this mechanism also operate in CD8+ T cells?
- 5) Iron restriction was associated with increased mitochondrial mass (Supplementary Fig. 1o). Were mitochondrial fission or fusion affected?
- 6) The data suggest that aspartate is trapped within mitochondria and becomes unavailable to support cell proliferation. Do the cells respond to this by increased mitophagy to liberate aspartate?
- 7) Superoxide was measured to assess mitochondrial function and health and the authors attribute its increase to low iron. Again, mitochondrial iron levels should be directly measured. Assessing mitochondrial respiration via Seahorse would provide direct functional clues.
- 8) "ACO2, which converts citrate to α -ketoglutarate". ACO2 converts citrate to isocitrate, which in turn is converted to α -ketoglutarate via isocitrate dehydrogenase. Thus, the decreased abundance of α -ketoglutarate could theoretically also be due to reduced isocitrate dehydrogenase activity. Direct enzymatic assays for aconitase, isocitrate dehydrogenase and glutamate dehydrogenase would address this issue.
- 9) "This dual dependency of KDMs for α -KG and iron mean that a decrease of α -KG availability under iron limitation may exert a double hit on KDM activity". A similar mechanism stabilizes HIF α ; why doesn't this work under iron restriction for HIF α ? Is it possible that iron is not limiting (since only tiny amounts are required), but rather oxidized to Fe³⁺ under pro-oxidant conditions? Can supplementation of ascorbate, a cofactor in the hydroxylation reaction, with or without dimethyl- α -KG rescue H3K27me3 levels? How did aspartate counteract the accumulation of the repressive histone mark H3K27me3 by iron deficient CD8+ T-cells?
- 10) "PPAT, the initiating enzyme of purine synthesis which has been predicted to be iron dependent was also reduced in iron scarcity (Fig. 5d)". This is at the level of protein abundance, not activity, which is opposite to what is observed for ETC and β -oxidation enzymes. Please, comment.
- 11) "Samhd1-KO CD8+ T-cells were less sensitive to iron scarcity in terms of a block on proliferation relative to wildtype cells (Supplementary Fig. 6b). This effect was modest, likely explained by the fact that SAMHD1-KO prevents dNTP breakdown but cannot rescue impaired nucleotide production". It is possible that the effect was modest because RNR is an iron-dependent enzyme.
- 12) "...aspartate supplementation did permit CD8+ T-cells to maintain high TFR1/CD71 expression in the lowest iron concentrations when expression typically begins to drop (Fig. 7e)". There is a typo (Fig. 6e instead of 7e). Why does TFR1 expression typically drop? This has been shown to occur under prolonged iron chelation via tristetraprolin (PMID 23102618) but this does not seem to be the case here.

Reviewer #3

(Remarks to the Author)

The work by The et al. proposes that iron deficient cells fail to proliferate due to increased CDKN1A expression without effects on viability. CD8 T cell activation is also impaired with KDM6, a histone demethylase, being iron and α -KG-dependent, thus maintaining high levels of H3K27me3. The authors describe adaptation to low iron environments including: higher expression of transferrin receptor TFR1, increased mitochondrial ROS, and lower abundance of TCA cycle metabolites with increased activity of PC and M+3 metabolites. However, aspartate levels are increased, but not sufficient to maintain proliferation. However, the authors conclude that addition of aspartate to cell culture partially restored cell proliferation rates and T cell function. There are interesting but somewhat disconnected observations presented in this study, that fall short to support the that "Iron deficiency causes aspartate-sensitive metabolic and proliferative dysfunction in CD8+ T cells. Given the specific and mechanistic nature of these studies, a revised version might be more appropriate for a more specialist journal.

To support the conclusions presented in this manuscript the authors should consider the following general and specific comments on the study (these are some examples and not an exhaustive list):

General comments:

Are these iron-deficient cells simply not activating? What is the expression of functional and activation markers in the absence of proliferation? The addition of more phenotypic and functional markers (CD69, CD25, PD-1, TNF, IFN γ etc) would strengthen the study. The reviewer only realised the authors had access to OTI mice when reading the methods. This means a lot of functional assays could have been shown throughout the manuscript.

Authors should consider running a comprehensive time-course to address some major issues: how long does it take for iron-sulfur clusters to be turned over, if they can recycle the iron? Could that in part rescue the phenotype and be important in short term vs long term cultures in the experimental design? The same goes for the carbon tracing experiments, they were static measurements at one timepoint, but the discussion was on changes in flux.

A very concerning statement is that aspartate increases glycolysis-derived and total ATP production, as well as perforin and IFN γ production in the context of iron depleted CD8 T cells. These are strong phenotypes but they don't seem to be iron-dependent. Additionally, statistics for figure 6h is missing so it's hard to tell whether there really is an improvement in IFN γ production.

Specific comments:

- 1- Were the iron concentrations chosen based on serum levels of levels in the media? Is there a reason why there isn't a condition with no iron? Does it impact survival?
- 2- Does the increase in TFR1 mean that the cells respond better once new iron becomes available or is this related to a change in the cell differentiation state? TFR1 had been linked to early-mid activation of T cells and is a marker of proliferating cells.
- 3- Fig Supp 1h should be better discussed. There are several genes identified that are altered but not on the protein level. Since the emphasis is on methylation status as a readout, protein alterations should be highlighted or mentioned, as it closely related to the phenotype, more than mRNA.
- 4- It might strengthen the argument in figure 1 if the authors mention that p53 is a regulator of cellular metabolism, as has been shown by many groups, including KH Vousden group.
- 5- Line 167: preferably use the term 'hypoxic-like response', since there is oxygen available in the model.
- 6- What is the role of the ETC/TCA cycle in HIF-1 α stabilisation during iron chelation/depletion? Does mTOR signaling which is low in iron deficient cells play a role? Some dioxygenases that are a-KG-dependent can be regulated by ROS too. Some more in-depth discussion could clarify the reasoning behind this panel.
- 7- In figure 1 f-m, could this phenotype be related to mitophagy? Is mitophagy an iron-dependent process?
- 8- In figure 2, authors are discussing mitochondrial alterations. Using an oligomycin control would strengthen the claims backed up by reference 5. Or even a Seahorse assay could answer the questions whether complex V is active in those cells or what is 'working' in the mitochondria.
- 9- In figure 3b the amount of glucose derived label is small. Same in other metabolites, such as serine. The errors on the unlabeled controls are concerning, authors could measure ¹³C lactate in the media too. Moreover, a 24hr labeling experiment should label more from ¹³C glucose. Line 231: the group of RG Jones has previously published works showing the serine can be imported, so it is not only a glycolytic intermediate. Lastly, when authors talk about decreased oxidative TCA cycle progression (line 258), they should show M+4 label from glutamine in the TCA and not only a-KG, it could be that glutamine-derived carbon is used more, so there is a relative drop in the a-KG label from glutamine. That could be higher metabolic flux. A time course experiment would help here in address labeling concerns and discuss metabolite flux.
- 10- Is there a direct activity assay (SDH, ACO₂, PCK2) the authors could use to show that there is lower enzymatic activity, along with reduced protein levels, as for e.g. PCK2? Iron deficiency led to increased PC activity and M+3 TCA cycle metabolites. Authors say that accumulation of aspartate is not for further nucleotide synthesis, but to reverse the TCA cycle and circumvent the iron-dependent enzymes SDH and ACO₂. Could this be just a collateral of increased PC activity? A lot of the conclusions in this figure (figure 5) come from potential utilisation of PC. A knockout experiment could show that this is really PC dependent with exogenous pyruvate. Also, please explain M+3 succinate and citrate in this context.
- 11- Figure 6: was the availability of dNTPs measured? This is key to support this figure, since catabolism and salvage could also drive carbons into central carbon metabolism via PNP.
- 12- How was the 40mM concentration of aspartate decided? Again, upregulation of CD71 could be a differentiation marker? In line 410 authors claim that aspartate can overcome P53 cell cycle inhibition despite elevated CDKN1A. This is a strong statement but is lacking functional data to back it up. The same is the case with line 420, when it is mentioned that aspartate does not attenuate mitochondrial dysfunction (CI, CII). This could be shown by blocking the complexes or running a Seahorse assay.

Comments on figures:

- 1- Figure 3b: the colour scheme is very confusing and the figure is hard to follow. It is not mentioned how the samples were normalized or whether we are looking at the gray or blue portions.
- 2- Figure 5g: the labeling does not match the text for the oxidative/reductive cycles.
- 3- Figure 6g is missing the colour legends.
- 4- Line 393 onwards: I believe you mean Fig 6 and not 7.

Reviewer #4

(Remarks to the Author)

Version 1:

Reviewer comments:

Reviewer #1

(Remarks to the Author)

I appreciate the author's efforts to improve the manuscript, but I have one comment. For Figure 4F, the author should

compare the H3K27Me3 levels on specific gene loci (such as CD25, CD71, CD98, IL-2, TNF, and IFN γ) under low versus high iron conditions.

Reviewer #2

(Remarks to the Author)

The revised manuscript is substantially improved and has addressed key issues raised by the reviewers.

Reviewer #3

(Remarks to the Author)

The diligent point by point answers and additional data provided in this rebuttal are appreciated, and the inclusion of the comprehensive activation/function panel related to T cell function strengthened the author's claims that iron deficiency dampens T cell function.

Unfortunately this reviewer still thinks that the often strong mechanistic claims made by the authors are not fully backed up by the data presented, both the original and additional data, resulting in limited enthusiasm for publication of this study as is in a high impact broad readership journal.

The interpretation of the metabolic data remains incomplete and in places is used to stretch the conclusions based on the limited assays performed.

The authors claim that there is lower oxidative TCA cycle activity and increased reductive TCA cycle, to circumvent iron dependent SDH and ACO2. The data provided for this falls short to support this point. The authors report "M+3 succinate from the reductive TCA cycle". The reductive TCA cycle flux is often reported as m+4 citrate from U13C GLN labelling, or m+3 malate and/or aspartate from U13C glucose labelling. M+3 succinate can be generated from the oxidative flux of the TCA as well, and as suggested, time course experiments would be helpful to unravel this. Long term labelling to achieve stable incorporation can cause scrambling of carbons in many intermediary metabolites making this interpretation difficult, especially when looking at directional incorporation such as directional fluxes in the TCA cycle (the removal of 'flux' to describe this data is appreciated).

The fact that cell counts can be a huge challenge to overcome when planning experiments with primary cells is not lost on this reviewer, but if the authors are not able to provide the data to support their conclusions, it becomes an assumption, leading to potential misinterpretations.

The authors did alter the statement that "aspartate supplementation confers a proliferative advantage was altered", to tone down the claimed that aspartate can overcome P53 cell cycle inhibition despite elevated CDKN1A. Similarly, authors removed the statement that aspartate does not attenuate mitochondrial dysfunction at CI and CII for being speculative, however, a Seahorse assay to validate this claim could have supported this with more mechanistic insight, which the authors preferred to leave out.

As the authors highlight in the rebuttal, aspartate supplementation and the subsequent phenotype of higher glycolytic rate and ATP production are not iron dependent observations. Conversely, the manuscript still claims (even in the title) that iron deficiency causes aspartate-sensitive metabolic and proliferative dysfunction in T cells.

Thus, although there are interesting changes that are associated with altered mitochondrial function in iron deficient T cells, the concern about mechanistic claims that are not fully backed up by the data remain.

Reviewer #4

(Remarks to the Author)

Version 2:

Reviewer comments:

Reviewer #1

(Remarks to the Author)

I am satisfied with the author's effort to improve the manuscript.

Reviewer #3

(Remarks to the Author)

I appreciate the detailed answers, and the text and title modification to better describe the data being shown.

I understand that revision rounds are exhausting when the story might already seem so clear to the authors, but it would have further strengthened the study if some additional supporting experiments would have been performed to further cement the metabolic findings.

This reviewer still struggles with the interpretation of the "reductive TCA" and especially the m+3 citrate label, since that could well be coming from oxidative TCA through PC activity, especially given the larger relative increase in the m+5 pools

compared to the m+3 pools in panel 5H. At the very least the authors should show the labeling in the full range of isotopomers (m+0 through m+6) of citrate in that context.

Reviewer #4

(Remarks to the Author)

Dear Reviewers,

We would like to thank you for taking the time to review our manuscript. Please find below a point-by-point response to your comments.

REVIEWER COMMENTS

Reviewer #1 (Remarks to the Author)

This study employs an *in vitro* model to investigate the influence of iron on CD8 T cell activation. The researcher's findings suggest that iron deficiency diminishes cell proliferation, modifies histone methylation status, and disrupts mitochondrial membrane integrity. Moreover, there is inadequate utilization of glucose and glutamine in the TCA cycle as revealed by the metabolic profile. The author impressively adopts a multi-omics approach to characterize epigenetic regulation, gene expression, protein abundance, and metabolite levels. Nevertheless, the manuscript lacks coherent organization and focus. Additionally, since this concept has not been tested using an *in vivo* or genetic model it reduces enthusiasm for this study.

Furthermore, the mechanism underlying reduced cell division due to iron deficiency remains unclear. Several concerns arise:

1) The researcher did not analyze any T-cell activation (CD44, CD69, and CD25) or functional markers until Figure 6. This makes it challenging to conclude the impact on T cell function. According to Figure 6, the absence of iron only decreases cell proliferation without affecting perforin and IFN γ levels. In the low holotransferrin group, less than 40% of cells were activated and divided; however, there was no change in overall perforin and IFN γ expression. This aspect is perplexing.

Our previous work (Fig S3E-G in Hepcidin-Mediated Hypoferremia Disrupts Immune Responses to Vaccination and Infection: Med) showed that *in vitro* activated CD8+ T-cells in low iron conditions have reduced CD98 and CD25 expression but no change in CD69 expression. In response to the reviewer's request, we have repeated and extended these findings, adding further functional data to this manuscript in Fig. 1d-j. We show CD25 and CD98 are substantially reduced by iron limitation, CD44 is not affected, IL-2 and TNF- α are induced by iron deficiency while IFN- γ is not significantly changed. These findings show that low iron interferes with the activation profile of OT-I T-cells so that proliferation is impaired, and expression of activation markers are variably affected with a general imbalance of effector functions. This is consistent with immediate signalling via TCR being relatively unaffected by low iron, but that subsequent downstream metabolic reprogramming and cell division is more profoundly influenced. Note that although on a per cell basis IFN- γ is unchanged and IL2 and TNF- α are increased by low iron, because cell division is strongly impaired (meaning that at the lowest iron concentration there are 64% fewer cells after 72 hours of culture (cell counts from Fig. 6b)) the overall effect of low iron availability at population level would be to inhibit a functional immune response, in line with our earlier *in vivo* findings (Hepcidin-Mediated Hypoferremia Disrupts Immune Responses to Vaccination and Infection: Med).

We have also included the following text:

"In line with previous work showing that in vivo iron deprivation dramatically reduces T-cell expansion⁵, in vitro iron limitation profoundly suppressed cellular proliferation, measured using cell trace violet (CTV; Fig. 1b-c). Iron deficiency also impaired expression of the activation marker, CD25, while having no effect on CD44 (Fig. 1d-e). Surface expression of the iron uptake receptor, TFR1/CD71, was increased in low iron conditions (Fig. 1f) consistent with iron limitation¹⁷ while the amino acid transporter, CD98 (SLC3A2:SLC7A5 heterodimer; LAT1), which is induced post-T-cell activation¹⁸, was significantly decreased in low iron conditions (Fig. 1g). In line with previous reports^{19,20}, the cytokine, IL-2, was induced during iron deprivation (Fig. 1h). TNF α was similarly induced with iron scarcity (Fig. 1i) while IFN- γ was unchanged (Fig. 1j). Therefore, in low iron conditions, stimulated OT-I T-cells acquire some though not all characteristics

of activation on a per-cell basis, and because of a failure to proliferate, overall, there are fewer competent effector T-cells when iron is restricted.”

Fig. 1d-j:

A significant number of cells fail to proliferate in the low iron conditions relative to high iron conditions, however these cells have activated and are transcriptionally distinct from naïve, resting cells (supplementary fig.1c). For example, in all experiments, cells activated under low iron conditions have elevated CD25 expression and size (measured as forward scatter; FSC) compared to resting cells maintained in a naïve like state using IL-7 (data added to supplementary Fig. 1a-b). This is consistent with the cells also having the capacity to express perforin and IFN γ expression. We have added the following text:

“Notably, the impaired proliferation and altered effector function observed under low iron conditions is not due to a failure to activate. Cells cultured in both low iron and iron replete conditions show robust upregulation of CD25 expression and increased cell size (measured as forward scatter; FSC) relative to naïve cells maintained in IL-7 (Supplementary Fig. 1a-b). Further, cells in low iron display a transcriptional profile distinct from naïve cells which is substantially more similar to cells in iron replete conditions (Supplementary Fig. 1c).”

Supplementary Fig. 1a-b:

2) The False Discovery Rate for Figure 1F should be provided by the author.

The adjusted P-values were calculated using the Benjamini-Hochberg procedure also known as the false discovery rate procedure. By definition, the adjusted P-values in this case are false discovery rate adjusted p-values. However, for clarity we have changed how this is annotated on the applicable plots (Fig. 1m and Supplementary Fig. 1l).

Fig. 1m:

Supplementary Fig. 1l:

3) An explanation is needed from the Author about why p53 regulated by iron only affects cell division without impacting survival measures.

We emphasize that we have not directly demonstrated activation of p53, only that the p53 regulated gene signature is induced at both the RNA and protein levels in low iron conditions. Indeed, p53 (encoded by *Trp53*) gene expression is unchanged during low iron conditions (see below, data not included in the manuscript). What we do show is that *Cdkn1a*, a key target downstream of p53 and a cell cycle suppressor is dramatically upregulated, and we suggest that this is at least partly responsible for the reduction in cell proliferation. We have altered the text (below) to make it evident that we have observed changes in genes associated with the p53 pathway rather than altered activity of P53 itself.

“In addition, gene set enrichment analysis (GSEA), conducted on both RNA-seq and protein-MS datasets also indicated altered expression of genes attributed to the p53 pathway (Fig. 1m, Supplementary Fig. 1l).”

“Altered activity of genes involved in the p53 pathway may therefore also contribute to the titratable impairment to cellular proliferation in low iron conditions (Fig. 1b-c).”

4) The inclusion of the “iron-sufficient” group is required in Figure 3.

The data shown for Fig. 3a is the relative amount of metabolite in the low iron condition relative to their matched iron sufficient control which is normalised to 1. Therefore, values greater than 1 show an increase in metabolite abundance in low iron relative to replete iron, whereas values less than 1 show a decrease in metabolite abundance in low iron relative to high iron.

5) Figure 4 illustrates that the general amount of H3K27me3 is comparable in IL-7 naive and iron-deficient conditions. To demonstrate the overlap of the H3K27me3 site between these two states, conducting CHIPseq is essential for the author.

Our focus has been to understand the differences between low and high iron states of activated OT-I T-cells. Therefore, and to align with the other experiments in the manuscript, we have completed ChIPmentation for H3K27me3 under high and low iron conditions. Similar to what we observe using flow cytometry, H3K27me3 histone marks accumulate under iron deprivation (Fig. 4e-f). Notably we observe that H3K27me3 marks accumulate across the genome. We cannot conclusively say that the H3K27me3 state of iron deficient CD8+ T-cells is similar to IL-7 supported naive cells, however, the genome wide accumulation of H3K27me3 rather than at specific loci suggests a general failure to remove H3K27me3 methylation from the previous naive state. An alternative or additional cause could be changes in methylation in the low iron state by methylases. We have included the following text:

“Using ChIPmentation, we confirmed that H3K27me3 levels were significantly elevated at TSSs in iron deficient CD8+ T-cells relative to iron replete (Fig. 4e-f). The genome wide increase in H3K27me3 during iron deficiency agrees with the concept that iron deprivation generally impairs the activity of KDM6 enzymes during T-cell activation. It is also possible that increased methylation by methylases could contribute to the H3K27me3 accumulation phenotype.”

Fig. 4e-f:

6.) Supplying α -ketoglutarate does not rescue phenotype needs clarification.

Lysine demethylase (KDM) enzymes are dependent on the availability of each of iron, α -ketoglutarate (α -KG) and oxygen. Thus, supplementation of α -KG may replenish depleted α -KG stocks in iron deficient OT-I T-cells, but KDM enzymes will still not be functional in the absence of iron, so for example α -KG supplementation did not affect H3K27me3 levels. Supplementation of α -KG did not enhance cell proliferation either. This has been clarified in the text:

“Of note, however, direct supplementation with cell-permeable dimethyl- α -KG failed to rescue H3K27me3 levels and cellular proliferation (Supplementary Fig. 5c-d) in iron deprived T-cells. This is consistent with KDM enzymes requiring all of iron, oxygen and α -KG. Thus, even if α -KG is no longer limiting, KDM enzymes will remain dysfunctional in the absence of sufficient iron cofactors.”

Reviewer #2 (Remarks to the Author)

This study investigates mechanisms by which iron deficiency impairs CD8+ T cell proliferation. The authors developed an elegant cell culture model for OT-I CD8+ T cells in the presence of variable concentrations of holo-transferrin. Iron limitation hampered cell proliferation without impacting cell viability. It also altered gene expression and disrupted mitochondrial functions. Iron-deficient CD8+ T cells had elevated levels of aspartate, a key mitochondria-derived metabolite. However, this aspartate wasn't utilized for nucleotide synthesis, likely due to trapping within depolarized mitochondria. Supplementation with exogenous aspartate, which directly accesses the cytosol, rescued clonal expansion of iron-deficient CD8+ T cells.

These data validate the importance of electron transport chain (ETC)-linked aspartate synthesis in cell proliferation, as proposed in references 35-36 and, furthermore, provide an important link to iron metabolism. The findings are novel, and

the experiments are rigorously designed and controlled. The conclusions are largely supported by the data, but there are issues requiring attention.

1) The induction of ETC and β -oxidation proteins, as well as of heme and Fe-S biosynthetic enzymes in iron-deficient cells is counterintuitive. Why would the cells waste so much energy when a critical co-factor is limiting? While the data show that limited iron supply triggers cellular responses to iron deficiency (Tfrc induction), there is no evidence for mitochondrial iron deficiency. Cytosolic and mitochondrial iron levels should be quantified.

Supplementary Fig. 3a-c includes analysis of data taken from Howden *et al* (Quantitative analysis of T cell proteomes and environmental sensors during T cell differentiation | Nature Immunology), a proteomics dataset of CD8+ T-cells activated for 0h, 24h and 6 days. This dataset does not include iron modulation. In this figure we are showing that heme and Fe-S cluster synthesis machinery is normally upregulated during T-cell activation: this information, although derived from the published datasets, has not been previously reported.

We did not observe significant changes in abundance of heme and Fe-S cluster proteins during iron deficiency in our proteomics data.

As the reviewer notes, we did observe increases in electron transport chain (ETC) and β -oxidation proteins in low iron conditions. We hypothesise that the cells are either trying to compensate for an energetic failure (previously observed by our group in iron deficient T-cells (Hepcidin-Mediated Hypoferremia Disrupts Immune Responses to Vaccination and Infection: Med) by increasing expression of the ETC machinery, or alternatively there is a mitochondrial clearance issue (please also see the response below to the specific comment on mitophagy) so that mitochondrial proteins relatively accumulate in iron deficient OT-I T-cells.

We are not currently aware of an accurate and sensitive enough method to accurately measure compartmentalised iron, including within mitochondria. Our group is currently developing a method for measuring cellular iron content by single cell inductively coupled plasma mass spectrometry (SC-ICP-MS) and have demonstrated a titratable decrease in cellular iron content with reduced iron availability (see below, data not included in this manuscript but available here: Rapid and precise quantification of lymphocyte iron content by single cell inductively coupled plasma mass spectrometry | bioRxiv), however, this method is still a long way off from being able to measure organellular iron content.

[figure redacted]

We examined the expression of the mitochondrial iron importers, mitoferrin 1 and 2 (Slc25a37 and Slc25a28 respectively), to try and understand whether mitochondrial iron import may be changing (see below, data not included in the manuscript). Mitoferrin 1 was decreased but not significantly, and mitoferrin 2 was not altered during iron scarcity.

2) “RNA expression changes observed during iron deficiency are largely due to transcriptional differences rather than alterations in RNA stability”. Does this apply to ETC and β -oxidation proteins, and heme and Fe-S biosynthetic enzymes?

We did not observe significant changes to ETC and B-ox proteins at the transcript level, only the proteomic level. This suggests that the regulation of these proteins is occurring post-transcriptionally. As mentioned above, heme and Fe-S cluster biosynthetic enzyme expression was not significantly different at either the transcript or protein level.

3) Is the induction of *Tfrc* under iron-restricted conditions likewise transcriptional? Did the proteomics analysis identify increased IRP2 levels in iron-deficient cells? This is a good marker of post-transcriptional responses to iron deficiency. Why is *Tfrc* not included among the list of upregulated genes/proteins in supplementary Fig. 1h?

Tfrc expression is regulated by both transcription and mRNA stability. The metabolic regulator, MYC, which is activated downstream of T-cell receptor stimulation, promotes *Tfrc* transcription which drives the dramatic increase in TFR1 protein expression in activated vs naïve T-cells (Data derived from Marchingo *et al*: Figures and data in Quantitative analysis of how Myc controls T cell proteomes and metabolic pathways during T cell activation | eLife, see below, data not included in the manuscript). *Tfrc* expression is also regulated post-transcriptionally by the iron regulatory proteins, ACO1 (IRP1) and IREB2 (IRP2). During iron limitation, ACO1 and IREB2 become activated and bind to the 3' end of the *Tfrc* mRNA promoting increased *Tfrc* mRNA stability and consequently increase TFR1 protein expression. H3K27ac and H3K4me3 are histone modifications typically associated with increased transcriptional activity. We observed that there were similar levels of H3K27ac and H3K4me3 at the *Tfrc* promoter between high and low iron conditions, consistent with similar accessibility and potential for transcription between iron conditions (data added to Supplementary Fig. 1f). However, there was significantly higher levels of *Tfrc* mRNA and protein under iron restriction (Fig. 1f and supplementary Fig. 1g-h). These findings indicate that the increase in *Tfrc* expression is due to increased mRNA stability (or other post-transcriptional effects) rather than increased transcription. However, we do acknowledge that these two histone markers (H3K27ac and H3K4me3) are not exhaustive and other modifications may be contributing to the observed effects. The following text has been added:

“In contrast, the iron uptake receptor, Tfrc, showed no observable difference in Tfrc promoter H3K4me3 or H3K27ac but a significant increase in mRNA and protein expression in low iron conditions (Supplementary Fig. 1f-h). These data are consistent with the canonical post-transcriptional upregulation of Tfrc via increased mRNA stability under iron limitation by the iron-response element (IRE)/iron response protein (IRP) system¹⁷.”

Supplementary Fig. 1f:

There were trends for both ACO1 and IREB2 towards increased protein expression in low iron conditions that did not reach statistical significance (see below, data not included in the manuscript). Further, work previously published from our group (Hepcidin-Mediated Hypoferremia Disrupts Immune Responses to Vaccination and Infection: Med) showed that in the absence of IRP proteins, activated T-cells do not fully upregulate TFR1 protein expression and fail to proliferate, indicating that IRP proteins are critical for TFR1 increase and for iron acquisition to support cell division. While we cannot conclusively state that the IRPs are responsible for the upregulation of *Tfrc* within T-cells in low iron conditions, our data is consistent with the accepted model of IRPs increasing *Tfrc* mRNA stability (rather than transcription) leading to increased TFR1 protein expression.

The list of upregulated genes/proteins in supplementary Fig. 1k is based on transcript thresholds of $\log_2|\text{FC}| > 1.5$ and FDR adjusted p-value < 0.05 and protein thresholds of $\log_2|\text{FC}| > 0.585$ ($|\text{FC}| > 1.5$) and p-value < 0.05 . While *Tfrc* mRNA expression is significantly different between iron conditions, it only has a $\log_2\text{FC} = 0.642$ ($\text{FC} = 1.6$), meaning that it does not reach the required threshold to be included in the analysis. Given that *Tfrc* upregulation at the RNA level is consistent across multiple repeats, we believe that this is a true phenomenon. However, we decided for stringency, to keep the fold change threshold at $\log_2|\text{FC}| > 1.5$. Note that TFR1 protein expression had a $\log_2\text{FC} = 0.597$ ($\text{FC} = 1.51$) and did make the threshold. This result is consistent with TFR1 protein being driven by both transcriptional and post-transcriptional regulation.

4) The cell cycle arrest observed in iron-deficient cells is attributed to CDKN1A induction. However, aspartate supplementation that rescued cell proliferation “only marginally reduced Cdkn1a expression in low iron conditions”. Iron deficiency is also known to affect other cell cycle check points; for instance, p21Cip1/Waf1 (PMID 17429006). Could this mechanism also operate in CD8+ T cells?

p21 is another name for *Cdkn1a* and so we certainly believe that iron deficiency is affecting this axis. This has been clarified in the text and we have added your requested reference:

“Consistent with this, expression of *Cdkn1a* (also known as p21, *Cip1* and *Waf1*), a p53 target and suppressor of the G1-S phase cell cycle transition was induced by iron deprivation at 24h, prior to the first cell division and continued to be upregulated as CD8+ T-cells entered their proliferative phase at 48h (Fig. 1n-p, Supplementary Fig. 1m). However, while cell division halted, cells remained viable at 48h (Fig. 1q). Notably, *Cdkn1a* is induced by iron chelation in cell lines²³.”

5) Iron restriction was associated with increased mitochondrial mass (Supplementary Fig. 1o). Were mitochondrial fission or fusion affected?

The key mitochondrial fusion genes, *Opa*, mitofusin 1 and 2 (*Mfn1* and *Mfn2*), and fission gene, *Dnml1* (also known as *Drp1*), are not significantly altered under iron limitation (see below, data not included in the manuscript). These genes were previously used as markers of mitochondrial dynamics in T-cells by this publication: Mitochondrial Dynamics Controls T Cell Fate through Metabolic Programming: Cell. For this reason, we did not follow up on this aspect of mitochondrial function and state.

Expression of mitochondrial fusion and fission genes. CD8⁺ T-cells were activated as described in Fig. 1a in a variety of holotransferrin concentrations. (a) mRNA (n=3) and (b) protein (n=4) expression of *Mfn1*. (c) mRNA (n=3) expression of *Mfn2*. (d) mRNA (n=3) and (e) protein (n=4) expression of *Opa1*. MFN1, MFN2 and OPA1 are involved in mitochondrial fusion. (f) mRNA (n=3) and (g) protein (n=4) expression of *Dnm1l* (DNMT1). DNMT1 (also known as DRP1) is involved in mitochondrial fission. Data are mean \pm SEM. Statistics are: (a, c, d, f) matched two-tailed t-tests (b, e, g) sampled matched one-way ANOVAs with the Geisser-Greenhouse correction.

6) The data suggest that aspartate is trapped within mitochondria and becomes unavailable to support cell proliferation. Do the cells respond to this by increased mitophagy to liberate aspartate?

We assessed the expression of key genes involved in autophagy, including the specific cargo adaptor proteins required for mitophagy, Bnip3l, Optn, Fundc1 and Tax1bp1 (see below, data not included in the manuscript). These genes were previously utilised as markers of autophagy including mitophagy by this pre-print Autophagy repression by antigen and cytokines shapes mitochondrial, migration and effector machinery in CD8 T cells | bioRxiv. We did not observe any significant shifts in general autophagy or mitochondria-specific adaptors required for mitophagy in iron deficient cells. Consequently, we did not further probe for differences in these processes at the cellular level.

7) Superoxide was measured to assess mitochondrial function and health and the authors attribute its increase to low iron. Again, mitochondrial iron levels should be directly measured. Assessing mitochondrial respiration via Seahorse would provide direct functional clues.

Work previously conducted by our lab (Hepcidin-Mediated Hypoferremia Disrupts Immune Responses to Vaccination and Infection: Med) using the same CD8⁺ T-cell iron deficiency model showed a reduction of total and mitochondrial ATP production but not glycolytic ATP production measured using the Seahorse ATP rate kit (data from *Frost et al, 2021* shown below). The mitochondrial ATP production rate is calculated using the oligomycin-sensitive oxygen consumption rate, therefore indicating that ATP coupled oxygen consumption, this data indicates that mitochondrial respiration is reduced in iron deficient conditions. Reduced ATP measured using cell titre glo was also observed in low iron conditions in this study.

Frost et al. 2021, Figure 4c:

[figure redacted]

As mentioned above, we do not have access to a method which can accurately measure organellar iron content. We have developed a method to measure cellular iron on a single cell basis via SC-ICP-MS and have shown in this pre-print (data above; Rapid and precise quantification of lymphocyte iron content by single cell inductively coupled plasma mass spectrometry | bioRxiv) that total cellular iron content titrates with iron availability.

8) “ACO2, which converts citrate to α-ketoglutarate”. ACO2 converts citrate to isocitrate, which in turn is converted to α-ketoglutarate via isocitrate dehydrogenase. Thus, the decreased abundance of α-ketoglutarate could theoretically also

be due to reduced isocitrate dehydrogenase activity. Direct enzymatic assays for aconitase, isocitrate dehydrogenase and glutamate dehydrogenase would address this issue.

We agree with the reviewer that this could also explain our findings. It was not possible for us to discriminate between citrate and isocitrate by MS analysis. We have therefore adjusted the text to clarify this point:

“ACO2 is an iron-dependent enzyme in the TCA cycle, which converts citrate to isocitrate. Isocitrate dehydrogenase 2 (IDH2) then converts isocitrate to α -ketoglutarate (α -KG). In agreement with reduced TCA cycle activity, decreased abundance of α -KG was observed in low-iron cells (Fig. 3a-b). Decreased α -KG M+3 isotopomers from ^{13}C -5-glutamine and a trend towards reduced α -KG M+2 isotopomers from ^{13}C -6-glucose supports reduced ACO2 and/or IDH2 activity (Fig. 3c, Supplementary Fig. 4e-g).”

We have also previously attempted enzymatic activity assays in CD8+ T-cells with our iron titration setup. Unfortunately, we have found that these assays have typically been designed to be used with cell lines or tissue samples where starting material is not limiting. Consequently, even with large numbers of *in vitro* activated CD8+ T-cells ($>1 \times 10^6$ cells), in our hands these assays lack the sensitivity to detect enzymatic activity in CD8+ T-cells, with signal only just above background, whereas they can easily detect activity in liver tissue (see below, data not included in the manuscript). Enzymatic activity has previously been detected in activated human CD4+ T-cells using similar enzymatic activity kits (complexes I, II, IV, V) Tumor-derived TGF- \$\beta\$ inhibits mitochondrial respiration to suppress IFN- \$\gamma\$ production by human CD4+ T cells | Science Signaling, however, up to 10×10^6 cell were required, cell numbers which are extremely difficult to acquire with our model, because OT-I T-cells do not proliferate in low iron conditions. Consequently, we could not use enzyme activity kits in our setting.

9) “This dual dependency of KDMs for α -KG and iron mean that a decrease of α -KG availability under iron limitation may exert a double hit on KDM activity”. A similar mechanism stabilizes HIF α ; why doesn't this work under iron restriction for HIF α ? Is it possible that iron is not limiting (since only tiny amounts are required), but rather oxidized to Fe $^{3+}$ under pro-oxidant conditions? Can supplementation of ascorbate, a cofactor in the hydroxylation reaction, with or without dimethyl- α -KG rescue H3K27me3 levels? How did aspartate counteract the accumulation of the repressive histone mark H3K27me3 by iron deficient CD8+ T-cells?

Thank you for these insightful questions. Different α -KG dependent enzymes have different affinities for iron; it has previously been reported that PHD2, which regulates Hif α stability, has an unexpectedly high affinity for iron (Hypoxia-inducible factor prolyl hydroxylase 2 has a high affinity for ferrous iron and 2-oxoglutarate - Molecular BioSystems (RSC Publishing)). In contrast to low iron availability, the iron chelator deferiprone stabilised HIF1 α (Supplementary fig. 2a-b). This is likely due to the fact that chelators have very high affinities for iron, and potentially compete for iron with even the highest affinity iron binding proteins, such as PHD2, whereas under iron limitation by decreasing Tf-iron availability, it may be that enzymes with lower affinities for iron are preferentially affected. For example, KDM3B and KDM3A were recently found to have Michaelis constants ($K_{M[\text{Fe}]}$) for iron of $K_{M[\text{Fe}]} = 100 \pm 30 \mu\text{M}$ and $K_{M[\text{Fe}]} = 2.6 \pm 0.8 \mu\text{M}$ respectively indicating that KDM3A has a much higher affinity for iron (Iron drives anabolic metabolism through active histone demethylation and mTORC1 | Nature Cell Biology). This same publication suggests that KDM3B, an H3K9me2 demethylase, senses iron deprivation via loss of its iron cofactor resulting in accumulation of repressive H3K9me2 and downregulation of

leucine import proteins. This publication gives precedence for the concept of iron sensing by differential affinity of 2-oxoglutarate dependent, iron dependent, and oxygen dependent enzymes for their iron cofactor. The following text has been included in the discussion:

“Increased H3K27me3 in iron deficient CD8+ T-cells is suggestive of KDM6 dysfunction. The inability of CD8+ T-cells to remodel their chromatin environment likely has implications for cellular differentiation. Moreover, suppressed levels of α -KG and/or iron may be capable of inhibiting iron and α -KG-dependent enzymes more generally including other KDM enzymes and the ten-eleven translocation (TET) DNA demethylases, as has been seen for KDM3B, which can act as an iron sensor influencing H3K9 methylation and mTORC1 activation⁶¹, and the PHD proteins which regulate HIF1 α ^{27,62}. However, iron scarcity did not stabilise HIF1 α or induce a proteomic response resembling that induced under hypoxic conditions. Different iron-dependent enzymes have been shown to have differential responses to iron inaccessibility, likely due to different iron binding affinities and expression levels⁶¹. For instance, KDM3B has a 40-fold lower affinity for iron than KDM3A ($KM_{[Fe]} = 100 \pm 30 \mu\text{M}$ and $KM_{[Fe]} = 2.6 \pm 0.8 \mu\text{M}$ respectively), providing rationale for why iron chelation results in accumulation of repressive H3K9me2 in a KDM3B, but not KDM3A, dependent manner⁶¹. Similarly the PHD proteins have very high affinities for iron⁶³, potentially explaining our result that iron chelators which potently suppress iron availability induce a HIF1 α response while iron deficiency caused by low extracellular transferrin-iron supply cannot. Thus, iron deprivation does not blanket inhibit all iron-dependent pathways and iron chelation does not fully recapitulate physiological iron-deficiency.”

The typical low iron condition we use has very minimal amounts of iron (0.001 mg/mL holotransferrin). Further, the upregulation of *Tfrc* expression (Fig. 1f, Supplementary Fig. 1g-h) agrees with sensing of a low iron state by the iron response protein (IRP), iron response element (IRE) system. As the reviewer suggests, it is possible that the oxidised state of the cell during iron deficiency (increased mROS and a higher NAD⁺/NADH ratio) may further impact the redox state of α -KG/iron/oxygen dependent dioxygenases, however we do not observe strong induction of an NRF2 antioxidant response in low iron conditions as measured via mRNA expression of the NRF2 target genes, *Gclc*, *Hmox1* and *Nqo1* (see below, data not included in the manuscript). This suggests that the increased oxidative state may be localised to the mitochondria, with increased mROS and induction of the mROS detoxifying protein, SOD2 (Fig. 2h and 2k). Further, ascorbate has been shown to have suppressive effects on CD4⁺ T-cells including reduced cytokine production (IL-17, IL-22 and IFN- γ) (A screen of Crohn's disease-associated microbial metabolites identifies ascorbate as a novel metabolic inhibitor of activated human T cells - *Mucosal Immunology*) suggesting that ascorbate supplementation under iron limitation may produce results that are difficult to interpret.

We propose that in the presence of exogenous aspartate, which can be utilised for downstream nucleotide production, the cells may no longer prioritise iron for mitochondrial substrate production, and this may liberate iron for use elsewhere in the cell, for instance in the nucleus by α -KG/iron/oxygen dependent dioxygenases. In new experiments, we have observed by LC-MS that aspartate increased the abundance of the purine and pyrimidine precursors, SAICAR and N-carbamoyl-aspartate (albeit not significantly) which were suppressed by iron deficiency. This data has been added as Fig. 6j-k with the following text:

“Aspartate treated CD8+ T-cells also displayed increased glycolytic and total ATP production (Supplementary Fig. 8i-k) and increased the abundance of the purine and pyrimidine precursors, SAICAR and N-carbamoyl-aspartate, albeit not significantly, suggesting that aspartate partially rescues nucleotide production, whilst consumption likely also increases in these more highly proliferating cells (Fig. 6j-k).”

Fig. 6j-k:

10) “PPAT, the initiating enzyme of purine synthesis which has been predicted to be iron dependent was also reduced in iron scarcity (Fig. 5d)”. This is at the level of protein abundance, not activity, which is opposite to what is observed for ETC and β -oxidation enzymes. Please, comment.

The reviewer makes a fair observation that some enzymes we predict are dysfunctional are increased in abundance while others are decreased in expression. We were trying to make the point that metabolic proteins are one of the key classes of proteins which are dysregulated during iron deficiency, and we apologise if this was not clear. Regarding PPAT, we don't know if iron levels are directly important for PPAT expression, however, by analysing data from the Howden et al. (Quantitative analysis of T cell proteomes and environmental sensors during T cell differentiation | Nature Immunology) dataset, we can see that T-cell activation dramatically induces PPAT expression at 24h and that this is a mTORC1 dependent process as rapamycin suppresses PPAT expression (see below, data not included in the manuscript). Thus, suppression of PPAT expression in low iron may be due to the reduced mTORC1 activity that occurs in low iron conditions (see Fig. 2b-c) and activity of the enzyme may be further compromised by lack of iron cofactor availability. While we cannot conclusively say that PPAT activity is reduced, we do observe a significant and dramatic reduction in the de novo purine synthesis intermediate, AICAR that is generated downstream of PPAT (Fig. 5c). So even if PPAT activity isn't impaired by iron deficiency, it does appear that purine synthesis is likely reduced overall. To address this comment, we have added the following text:

“PPAT, the initiating enzyme of purine synthesis which has been predicted to be iron-dependent⁷ was also reduced in iron scarcity (Fig. 5d). However, whether PPAT enzymatic activity is reduced is unclear.”

11) “Samhd1-KO CD8+ T-cells were less sensitive to iron scarcity in terms of a block on proliferation relative to wildtype cells (Supplementary Fig. 6b). This effect was modest, likely explained by the fact that SAMHD1-KO prevents dNTP breakdown but cannot rescue impaired nucleotide production”. It is possible that the effect was modest because RNR is an iron-dependent enzyme.

That is correct, we predict that the combined effect of impaired purine/pyrimidine synthesis and RNR activity would abrogate the SAMHD1 rescue. We have added a mention of RNR in the text to make this clearer:

“This effect was modest, likely explained by the fact that SAMHD1-KO prevents dNTP breakdown but cannot rescue impaired nucleotide production or iron-dependent RNR activity.”

12) “...aspartate supplementation did permit CD8+ T-cells to maintain high TFR1/CD71 expression in the lowest iron concentrations when expression typically begins to drop (Fig. 7e)”. There is a typo (Fig. 6e instead of 7e). Why does TFR1 expression typically drop? This has been shown to occur under prolonged iron chelation via tristetraprolin (PMID 23102618) but this does not seem to be the case here.

Thank you for spotting the typo, which we have fixed. We routinely observe that TFR1 expression drops at the lowest iron concentrations similar to the setting of prolonged iron starvation in the study you mention. It is certainly possible that TTP may be activated within our most iron deprived cells and may act against the classical IRP/IRE axis to try and conserve iron when exogenous iron is most limiting. Alternatively, given mTORC1 is known to regulate translational efficiency, TFR1 protein synthesis may be reduced due to the observed mTORC1 inhibition as evidenced by a reduction in its downstream target pS6 (Fig. 2b-c). We have added the text:

“TFR1/CD71 levels may drop in the lowest iron conditions due to induction of tristetraprolin (TTP), a protein previously shown to suppress TFR1 expression under prolonged iron chelation⁵⁵ or may be due to decreased TFR1 protein synthesis under mTORC1 suppression (Fig. 2b-c).”

Reviewer #3 (Remarks to the Author):

The work by The et al. proposes that iron deficient cells fail to proliferate due to increased CDKN1A expression without effects on viability. CD8 T cell activation is also impaired with KDM6, a histone demethylase, being iron and a-KG-dependent, thus maintaining high levels of H3K27me3. The authors describe adaptation to low iron environments including: higher expression of transferrin receptor TFR1, increased mitochondrial ROS, and lower abundance of TCA cycle metabolites with increased activity of PC and M+3 metabolites. However, aspartate levels are increased, but not sufficient to maintain proliferation. However, the authors conclude that addition of aspartate to cell culture partially restored cell proliferation rates and T cell function. There are interesting but somewhat disconnected observations presented in this study, that fall short to support the that “Iron deficiency causes aspartate-sensitive metabolic and proliferative dysfunction in CD8+ T cells. Given the specific and mechanistic nature of these studies, a revised version might be more appropriate for a more specialist journal.

To support the conclusions presented in this manuscript the authors should consider the following general and specific comments on the study (these are some examples and not an exhaustive list):

General comments:

Are these iron-deficient cells simply not activating?

A significant number of cells fail to proliferate in the low iron conditions relative to high iron conditions, however these cells have activated and are transcriptionally distinct from naïve, resting cells (supplementary fig.1c). For example, in all experiments, cells activated under low iron conditions have elevated CD25 expression and size (measured as forward scatter; FSC) compared to resting cells maintained in a naïve like state using IL-7 (data added to supplementary Fig. 1a-b). This is consistent with the cells also having the capacity to express perforin and IFN γ expression. We have added the following text:

“Notably, the impaired proliferation and altered effector function observed under low iron conditions is not due to a failure to activate. Cells cultured in both low iron and iron replete conditions show robust upregulation of CD25 expression and increased cell size (measured as forward scatter; FSC) relative to naïve cells maintained in IL-7 (Supplementary Fig. 1a-b). Further, cells in low iron display a transcriptional profile distinct from naïve cells which is substantially more similar to cells in iron replete conditions (Supplementary Fig. 1c).”

Supplementary Fig. 1a-b:

What is the expression of functional and activation markers in the absence of proliferation? The addition of more phenotypic and functional markers (CD69, CD25, PD-1, TNF, IFN γ etc) would strengthen the study. The reviewer only realised the authors had access to OTI mice when reading the methods. This means a lot of functional assays could have been shown throughout the manuscript.

Our previous work (Fig S3E-G in Hepcidin-Mediated Hypoferremia Disrupts Immune Responses to Vaccination and Infection: Med) showed that *in vitro* activated CD8+ T-cells in low iron conditions have reduced CD98 and CD25 expression but no change in CD69 expression. In response to the reviewer’s request, we have repeated and extended these findings, adding further functional data to this manuscript in Fig. 1d-j. We show CD25 and CD98 are substantially reduced by iron limitation, CD44 is not affected, IL-2 and TNF- α are induced by iron deficiency while IFN- γ is not significantly changed. These findings show that low iron interferes with the activation profile of OT-I T-cells so that proliferation is impaired, and expression of activation markers are variably affected with a general imbalance of effector functions. This is consistent with immediate signalling via TCR being relatively unaffected by low iron, but that subsequent downstream metabolic reprogramming and cell division is more profoundly influenced. Note that although on a per cell basis IFN- γ is unchanged and IL2 and TNF- α are increased by low iron, because cell division is strongly impaired (meaning that at the lowest iron concentration there are 64% fewer cells after 72 hours of culture (cell counts from Fig. 6b)) the overall effect of low iron availability at population level would be to inhibit a functional immune response, in line with our earlier *in vivo* findings (Hepcidin-Mediated Hypoferremia Disrupts Immune Responses to Vaccination and Infection: Med).

We have also included the following text:

“In line with previous work showing that *in vivo* iron deprivation dramatically reduces T-cell expansion⁵, *in vitro* iron limitation profoundly suppressed cellular proliferation, measured using cell trace violet (CTV; Fig. 1b-c). Iron deficiency also impaired expression of the activation marker, CD25, while having no effect on CD44 (Fig. 1d-e). Surface expression of the iron uptake receptor, TFR1/CD71, was increased in low iron conditions (Fig. 1f) consistent with iron limitation¹⁷ while the amino acid transporter, CD98 (SLC3A2:SLC7A5 heterodimer; LAT1), which is induced post-T-cell activation¹⁸, was significantly decreased in low iron conditions (Fig. 1g). In line with previous reports^{19,20}, the cytokine, IL-2, was induced during iron deprivation (Fig. 1h). TNF α was similarly induced with iron scarcity (Fig. 1i) while IFN- γ was unchanged (Fig. 1j). Therefore, in low iron conditions, stimulated OT-I T-cells acquire some though not all characteristics of activation on a per-cell basis, and because of a failure to proliferate, overall, there are fewer competent effector T-cells when iron is restricted.”

Fig. 1d-j:

Authors should consider running a comprehensive time-course to address some major issues: how long does it take for iron-sulfur clusters to be turned over, if they can recycle the iron? Could that in part rescue the phenotype and be important in short term vs long term cultures in the experimental design?

Fe-S cluster turnover is a very complex process to study. Using data from the Howden *et al* dataset (Quantitative analysis of T cell proteomes and environmental sensors during T cell differentiation | Nature Immunology), we previously showed that 60 different Fe-S cluster binding proteins are detectable in T-cells (Frontiers | Analysis of Iron and Iron-Interacting Protein Dynamics During T-Cell Activation). Each of these 60 Fe-S cluster binding proteins will have different affinities for iron and different rates of protein turnover. Further, Fe-S cluster distribution to their recipient proteins is highly dependent on Fe-S cluster chaperone proteins and it is still unclear which chaperones are responsible for furnishing all Fe-S cluster binding proteins. With this context in mind, we believe that study of Fe-S cluster turnover goes beyond the scope of our study. Further, due to the dramatic increase in predicted iron needs of activating T-cells and the dilution of iron between daughter cells as T-cells divide (as described here Frontiers | Analysis of Iron and Iron-Interacting Protein Dynamics During T-Cell Activation), liberation of iron from Fe-S cluster recycling is unlikely to provide sufficient iron for cellular function.

The same goes for the carbon tracing experiments, they were static measurements at one timepoint, but the discussion was on changes in flux.

We chose to conduct our carbon tracing experiments at 48h because this is the time point where we start to see substantial differences in cell state between high and low iron conditions without compromising viability. At 24h, the

cells have not divided yet and consequently do not show differences in proliferation. Further, we only observe a slight trend towards increased *Tfrc* expression at 24h (see below, data not included in manuscript) suggesting that prior to the first division (and the dilution of iron stores) cells do not experience significant cytosolic iron deprivation. Further, we found that in our model we required 24h of ¹³C6-glucose or ¹³C5-glutamine labelling to get sufficient label into mitochondrial metabolism thus limiting the time points we are able to query. We have also found that we require around 2x10⁶ CD8+ T-cells to get decent metabolite detection in our metabolite-MS facility. Consequently, culturing sufficient cells to run metabolite-MS with labelling at multiple matched iron conditions and time points is not very feasible (because OT-I T-cells do not proliferate well in low iron) and we believe we have conducted the labelling experiments as optimally as possible. However, appreciate the reviewer’s point that we have not conducted an experiment which can formally address flux and have thus removed the word “flux” from our manuscript.

A very concerning statement is that aspartate increases glycolysis-derived and total ATP production, as well as perforin and IFN γ production in the context of iron depleted CD8 T cells. These are strong phenotypes but they don’t seem to be iron-dependent. Additionally, statistics for figure 6h is missing so it’s hard to tell whether there really is an improvement in IFN γ production.

We agree that the aspartate related phenotypes the reviewer mentions are likely iron independent and we were not intending to suggest this was the case; indeed the point is that they are iron-independent. We apologise if this was unclear. Iron dependent aspartate effects include: cellular division, TFR1/CD71 MFI, H3K27me3 MFI and the NAD⁺/NADH ratio. Meanwhile, aspartate has effects on the following independent of iron: CD25 MFI, perforin MFI, IFN γ MFI, pS6 MFI, transcriptional suppression of interferon response genes and ATP production. This is consistent with a major role for aspartate in cellular metabolism as described for cancer cell lines (An Essential Role of the Mitochondrial Electron Transport Chain in Cell Proliferation Is to Enable Aspartate Synthesis: Cell and Supporting Aspartate Biosynthesis Is an Essential Function of Respiration in Proliferating Cells: Cell). Accordingly, we have edited the text

“Independent of iron, aspartate supplementation also promoted CD8+ T cell expression of the activation marker, CD25, the cytolytic molecule, perforin, and the cytokine, IFN- γ , indicating its availability within the cytosol supports multiple cellular processes beyond proliferation (Fig. 6f-h).”

Also, the statistics for Fig. 6h are listed at the bottom of the sub-figure with the p-values for the iron effect, aspartate effect and the interaction effect.

Specific comments:

1- Were the iron concentrations chosen based on serum levels of levels in the media? Is there a reason why there isn’t a condition with no iron? Does it impact survival?

The highest concentration of 0.625 mg/mL holotransferrin (approximately 15.6 μ mol/L of iron) was calculated to be roughly reflective of the levels of iron physiologically found in human sera (14-32 μ mol/L of iron). Meanwhile, the low holotransferrin concentrations of 0.001 mg/mL are much lower than that found in plasma but are reflective of the iron levels in the lymphatics which have been found to be almost negligible (Lymph protects metastasizing melanoma cells

from ferroptosis | Nature). Tissue iron availability may be even lower in contexts such as of the tumour microenvironment in the presence of iron demanding tumour cells (Hepcidin sequesters iron to sustain nucleotide metabolism and mitochondrial function in colorectal cancer epithelial cells | Nature Metabolism). Thus, the range of iron availability used likely does span physiological ranges. We have included the following text justifying our decision:

“0.625 mg/mL holotransferrin (15.6 $\mu\text{mol/L}$ of iron) is representative of the levels of iron found in human sera (14-32 $\mu\text{mol/L}$ of iron)¹⁵. Meanwhile 0.001 mg/mL holotransferrin is much lower than levels found in plasma but may be reflective of the almost negligible levels of iron detected in lymphatic fluid¹⁶.”

We did not include an iron free condition as we know that the 0.001 mg/mL holotransferrin condition is already very iron limiting. At iron concentrations lower than this viability does start to decrease. Crucially, viability remains high at the 0.001 mg/mL condition at 48h making it a nice concentration to work with as we can compare cells between conditions without the confounder of a large dying population within the culture.

2- Does the increase in TFR1 mean that the cells respond better once new iron becomes available or is this related to a change in the cell differentiation state? TFR1 had been linked to early-mid activation of T cells and is a marker of proliferating cells.

Tfrc expression in the context of iron deficiency cannot be interpreted purely as an activation marker in activated T-cells. Post-T-cell stimulation, *Tfrc* expression is induced by MYC to increase iron uptake, permitting the acquisition of sufficient iron for T-cell function (Data derived from Marchingo *et al*: Figures and data in Quantitative analysis of how Myc controls T cell proteomes and metabolic pathways during T cell activation | eLife, see below, data not included in the manuscript). However, *Tfrc* expression in activated T-cells is also modulated by iron availability and the iron response protein (IRP)/iron response element (IRE) system. Thus, in our culture conditions, we see an increase of TFR1 in response to T-cell activation, with an additional boost in TFR1 expression by iron deprivation. T-cells initially cultured in high iron but moved to low iron conditions upregulate TFR1 substantially within 24h (see below, data not included in manuscript). Conversely, T-cells initially cultured in low iron for 48h but transferred to high iron conditions reduce TFR1 expression within 24h. This data indicates that the difference in TFR1 expression between cells cultured in low vs high iron conditions is due to iron status rather than activation state. Further, cells moved from low to high iron conditions do show an increase in division relative to cells maintained in the low iron condition.

3- Fig Supp 1h should be better discussed. There are several genes identified that are altered but not on the protein level. Since the emphasis is on methylation status as a readout, protein alterations should be highlighted or mentioned, as it closely related to the phenotype, more than mRNA.

The table in Supplementary Fig. 1h (now Supplementary Fig. 1k) shows the genes which are both differentially expressed at the mRNA level using the thresholds $\log_2|FC| > 1.5$ and $FDR < 0.05$ and the protein level using the thresholds of $\log_2|FC| > 0.585$ ($|FC| > 1.5$) and $p\text{-value} < 0.05$. These are also the genes shown in red in Fig. 1l. Methylation state is not considered in this table.

We have predominantly highlighted protein changes in the main figures (Fig. 1-2) over RNA changes which are predominantly shown in supplementary figures (Supplementary Fig. 1) as we agree that protein changes more accurately reflect the cell state over transcript changes. We acknowledge that there are many changes which occur at the protein level, for instance increased expression of ETC and beta-oxidation proteins (Fig. 2d), which do not change at the transcript level.

4- It might strengthen the argument in figure 1 if the authors mention that p53 is a regulator of cellular metabolism, as has been shown by many groups, including KH Vousden group.

Thank you, we have implemented your recommendation as follows:

“p53 has also been shown to regulate cellular metabolism with effects including glycolytic suppression and mitochondrial maintenance¹⁵.”

5- Line 167: preferably use the term ‘hypoxic-like response’, since there is oxygen available in the model.

Thank you, this has been adjusted accordingly:

“Low iron availability induces a response distinctive from either hypoxia or iron chelation”

6- What is the role of the ETC/TCA cycle in HIF-1a stabilisation during iron chelation/depletion? Does mTOR signaling which is low in iron deficient cells play a role?

We observed treatment with iron chelator stabilised HIF-1a, but that media iron depletion did not result in HIF1a stabilisation (Supplementary Fig. 2a-b). Similarly, we analysed the publicly available GEO RNAseq dataset GSE84702, which features Th1 T-cells treated with the iron chelator CPX. CPX induced a very strong transcriptional response (>3000 differentially expressed genes) that is distinctive from the response to low iron availability we observed in OT-I T-cells (data added to Supplementary Fig. 2d-g). We posit that low extracellular iron availability in cell culture media induces a cellular state more reflective of physiological iron deprivation than high affinity iron chelators that can be cell permeant (like CPX) and which can lead to HIF1a stabilisation. Consequently, we have focused on low iron media for the majority of the manuscript. Given that we only see HIF1a stabilisation with iron chelators, we believe that the role of the ETC/TCA

cycle during HIF α stabilisation (as occurs during iron chelation but not in low iron media) goes beyond the remit of this manuscript. We have added the following text:

“Given the discrepancy between how HIF1 α responded in CD8+ T-cells under iron chelation and iron deficiency we were interested to understand whether the difference in cellular response extended beyond HIF1 α . We processed raw RNA-seq data from Wang et al²⁹ (not analysed in their manuscript) of Th1 polarised CD4+ T-cells treated with CPX iron chelator or dimethyl sulfoxide (DMSO) control for 4 hours. CPX treated cells showed differential expression of 3523 genes which is approximately 18-fold higher than the 193 differentially expressed genes identified in our model of low iron media (Supplementary Fig. 2d). CPX also failed to upregulate *Tfrc* (Supplementary Fig. 2e), suggesting that the IRP-IRE iron signalling system is no longer intact. While CPX iron chelation suppressed pathways associated with cell cycle (G2M checkpoint) and metabolic reprogramming (MYC targets; Supplementary Fig. 2f) similar to iron deficiency, only 45 genes were found to be mutually differentially expressed (Supplementary Fig. 2g). The extreme changes in transcriptional profile observed with only 4 hours of CPX treatment relative to our 48-hour culture model in low iron media and the failure to upregulate *Tfrc* suggests that the biochemical impacts of chelators can be extremely non-physiological, especially if given at high doses.”

Supplementary Fig. 2d-g:

We do observe reduced mTORC1 signalling in low iron conditions (Fig. 2b-c) in addition to increased ETC proteins (Fig. 2d) but reduced TCA cycle metabolism (Fig. 3). mTORC1 is important for many metabolic processes including glycolysis and mitochondrial biogenesis to support the TCA cycle and ETC (mTOR, metabolism, and the regulation of T-cell differentiation and function - Waickman - 2012 - Immunological Reviews - Wiley Online Library). In CD8+ T-cells,

mTORC1 inhibition with rapamycin reduces the expression of ribosomal and mitochondrial proteins (Quantitative analysis of T cell proteomes and environmental sensors during T cell differentiation | Nature Immunology). While we do see a reduction of ribosomal proteins under iron starvation, we conversely observe accumulation of ETC proteins. This suggests that while mTORC1 may cause some of the metabolic effects observed under iron deprivation, it is not completely responsible for the overall phenotype. We have added the following text to the discussion:

“Iron deficiency induced profound alterations to the transcriptional and proteomic landscape with significant perturbations in metabolic gene expression, including suppressed mTORC1 pathway associated genes and elevated OXPHOS and β -oxidation proteins. mTORC1 is essential for T-cell metabolic reprogramming downstream of TCR stimulation for ATP and biosynthetic substrate generation required for activation⁶⁰. Iron deficiency partially recapitulates the phenotype of mTORC1 inhibited CD8+ T-cells which feature reduced expression of ribosomal proteins¹⁴. However, in contrast to the decline in mitochondrial proteins observed with rapamycin¹⁴, we instead saw increased mitochondrial proteins indicating that iron deficiency’s impact on metabolism is not simply due to suppression of mTORC1. We hypothesise that accumulation of mitochondrial proteins in low iron conditions could be due to augmented generation of mitochondrial proteins to compensate for suppressed mitochondrial activity or impaired clearance by proteolysis or mitophagy.”

Some dioxygenases that are a-KG-dependent can be regulated by ROS too. Some more in-depth discussion could clarify the reasoning behind this panel.

We acknowledge that other mechanisms may explain the accumulation of H3K27me3 that we observe beyond substrate unavailability. For instance, as the reviewer suggests, ROS, may also inhibit KDMs. We have added the following text to address this:

“While we hypothesise that KDM6 impairment occurs due to iron deficiency coupled to suppressed α -KG production, KDM6 activity may be impaired by other mechanisms. For instance, ROS, which are induced by iron deficiency (Fig. 2h) have been shown to inhibit other iron-dependent, oxygen dependent and α -KG dependent dioxygenases such as the PHD proteins which regulate HIF1 α ⁴¹.”

7- In figure 1 f-m, could this phenotype be related to mitophagy? Is mitophagy an iron-dependent process?

It is possible that the accumulation of mitochondrial proteins may be due to defects in mitophagy. We assessed the expression of key genes involved in autophagy, including the specific cargo adaptor proteins required for mitophagy, Bnip3l, Optn, Fundc1 and Tax1bp1 (see below, data not included in the manuscript). These genes were previously utilised as markers of autophagy including mitophagy by this pre-print: Autophagy repression by antigen and cytokines shapes mitochondrial, migration and effector machinery in CD8 T cells | bioRxiv. We did not observe any significant shifts in general autophagy or mitochondrial specific adaptors required for mitophagy in iron deficient cells. Consequently, we did not follow up on particular angle.

We recently summarised the role of almost all iron interacting proteins in this review: Why cells need iron: a compendium of iron utilisation: Trends in Endocrinology & Metabolism. To our knowledge, mitophagy does not involve any iron dependent proteins.

8- In figure 2, authors are discussing mitochondrial alterations. Using an oligomycin control would strengthen the claims backed up by reference 5. Or even a Seahorse assay could answer the questions whether complex V is active in those cells or what is 'working' in the mitochondria.

Work previously conducted by our lab (Hepcidin-Mediated Hypoferremia Disrupts Immune Responses to Vaccination and Infection: Med) using the same CD8+ T-cell iron deficiency model showed a reduction of total and mitochondrial ATP production but not glycolytic ATP production measured using the Seahorse ATP rate kit (data from *Frost et al, 2021* shown below). The mitochondrial ATP production rate is calculated using the oligomycin-sensitive oxygen consumption rate, therefore indicating that ATP coupled oxygen consumption, this data indicates that mitochondrial respiration is reduced in iron deficient conditions. Reduced ATP measured using cell titre glo was also observed in low iron conditions in this study.

Frost et al. 2021, Figure 4c:

[figure redacted]

9- In figure 3b the amount of glucose derived label is small. Same in other metabolites, such as serine. The errors on the unlabeled controls are concerning, authors could measure ¹³C lactate in the media too. Moreover, a 24hr labeling experiment should label more from ¹³C glucose.

The data in Fig. 3b only shows data from ¹³C₅-glutamine tracing (not ¹³C₆-glucose). As would be expected given that glutamine enters the TCA cycle via glutamate rather than glycolysis, we do not observe large amounts of label in

metabolites derived from glycolytic intermediates such serine, alanine and lactate. To avoid confusion, we have removed this data. The figure now looks like this:

Fig. 3b:

We have also now included the supernatant lactate labelling from the ¹³C₆-glucose tracing experiment in Supplementary Fig. 4b. As you can see, there is no significant difference in the total amount of lactate between iron conditions suggesting no change in glycolysis with iron deficiency. If anything, we observe a trend towards reduced secreted lactate in the low iron condition suggestive of a reduced metabolic state. Further, we see 80% of lactate is labelled from ¹³C₆-glucose indicating that our tracer is successfully being incorporated into metabolism (added as Supplementary Fig. 4c). We have included the following text:

“Furthermore, there were no significant changes in the abundance of total lactate or the fraction of lactate labelled from ¹³C₆-glucose in the supernatants of iron deprived CD8+ T-cells (Supplementary Fig. 4b-c) together suggesting glycolytic activity was preserved under low iron conditions.”

Supplementary Fig. 4b-c:

Line 231: the group of RG Jones has previously published works showing the serine can be imported, so it is not only a glycolytic intermediate.

We have revised the sentence to:

“Here, minimal changes in overall abundance of the glycolytic metabolites, pyruvate and lactate, or the amino acids which may be derived from glycolytic intermediates or imported (alanine, serine, and glycine) were observed (Fig. 3a).”

Lastly, when authors talk about decreased oxidative TCA cycle progression (line 258), they should show M+4 label from glutamine in the TCA and not only a-KG, it could be that glutamine-derived carbon is used more, so there is a relative drop in the a-KG label from glutamine. That could be higher metabolic flux. A time course experiment would help here in address labeling concerns and discuss metabolite flux.

We have added the isotopomer abundance data for the $^{13}\text{C}_5$ -glutamine tracing for succinate, fumarate and malate to Fig. 3c-f (see below). As you can see, we observe a significant decrease in M+4 fumarate and M+4 malate in low iron conditions (and a non-significant decrease in succinate) during $^{13}\text{C}_5$ -glutamine tracing, further suggesting that there is reduced entry of $^{13}\text{C}_5$ -glutamine derived carbon into the TCA cycle. We also observed a reduction in M+4 aspartate derived from $^{13}\text{C}_5$ -glutamine further suggesting reduced glutamine utilisation and suppressed TCA cycle progression. We have added the following text:

“Decreased M+4 ^{13}C -labelling from $^{13}\text{C}_5$ -glutamine into fumarate and malate also supports this (Fig. 3d-f).”

As we discussed above to your previous point, 48h was chosen as a timepoint at which an iron deficiency phenotype was observable but prior to any large consequences for cellular division. Further, issues with producing sufficient cell numbers and the required tracing time has made doing a time course experiment extremely challenging.

Fig. 3c-f

10- Is there a direct activity assay (SDH, ACO2, PCK2) the authors could use to show that there is lower enzymatic activity, along with reduced protein levels, as for e.g. PCK2? Iron deficiency led to increased PC activity and M+3 TCA cycle metabolites. Authors say that accumulation of aspartate is not for further nucleotide synthesis, but to reverse the TCA cycle and circumvent the iron-dependent enzymes SDH and ACO2. Could this be just a collateral of increased PC activity? A lot of the conclusions in this figure (figure 5) come from potential utilisation of PC. A knockout experiment

could show that this is really PC dependent with exogenous pyruvate. Also, please explain M+3 succinate and citrate in this context.

We have previously attempted enzymatic activity assays in CD8+ T-cells with our iron titration setup. Unfortunately, we have found that these assays have typically been designed to be used with cell lines or tissue samples where starting material is not limiting. Consequently, even with large numbers of *in vitro* activated CD8+ T-cells ($>1 \times 10^6$ cells), we have found that these assays lack the sensitivity to detect enzymatic activity in CD8+ T-cells, with signal only just above background, although they easily detect activity in liver tissue (e.g. see below for Complex I activity, data not included in the manuscript). Enzymatic activity has previously been detected in activated human CD4+ T-cells using similar enzymatic activity kits (complexes I, II, IV, V) (Tumor-derived TGF- \$\beta\$ inhibits mitochondrial respiration to suppress IFN- \$\gamma\$ production by human CD4+ T cells | Science Signaling), however, up to 10×10^6 cell were required, cell numbers which are extremely difficult to acquire with our model because OT-I T-cells do not proliferate in low iron conditions. Consequently, we could not use enzyme activity kits in our setting.

We acknowledge that it is possible that increased PC activity and the reductive TCA cycle may also be acting to promote nucleotide production even if the result is futile. We have adjusted the text accordingly:

“This suggests that under iron limiting conditions, CD8+ T-cells may potentially utilise PC and the reductive TCA cycle to replenish the depleted metabolites fumarate and malate, circumventing the use of the iron-dependent enzymes, ACO2 and SDH. Alternatively, increased PC activity may be a consequence of a futile attempt to increase nucleotide production via aspartate.”

We agree that experiments conducted with a PC knockout would be informative, however, we do not believe that conclusively showing that iron deficiency results in a higher dependency on PC over PDH is essential for the overall message of our manuscript.

M+3 succinate would occur from reductive flux of the TCA cycle. M+5 citrate occurs when an M+3 oxaloacetate derived from PC activity combines with an M+2 acetyl-CoA derived from PDH whereas M+3 citrate would occur if an M+3 oxaloacetate combined with an unlabelled acetyl-CoA, for instance an acetyl-CoA derived from fatty acid beta-oxidation rather than glycolysis.

11- Figure 6: was the availability of dNTPs measured? This is key to support this figure, since catabolism and salvage could also drive carbons into central carbon metabolism via PNP.

Via LCMS we have now detected the pyrimidine dNTPs, dTTP, dTMP and dCTP, as well as purine dATP, but not dGTP. Our data indicate that dTTP abundance is consistently decreased in low iron conditions, with dCTP and dATP also trending to be decreased but dTMP abundance being overall similar (with high variation) in both conditions (data added as Supplementary Fig. 6c).

Supplementary Fig. 6c:

In addition, we have interrogated PNP expression in our transcriptomic/proteomic dataset and found that while *Pnp* mRNA is slightly reduced in low iron, PNP protein levels are not affected (see below, data not included in the manuscript). While not conclusive, the lack of change in PNP expression suggests that salvage may not be affected by iron deficiency and that nucleotide shifts are more likely to be due to the reduction in de novo nucleotide precursors. However, we have added the following text to indicate that salvage may also be playing a role:

“Consistent with these observations, certain nucleotides and dNTPs were decreased during iron-deficiency (Supplementary Fig. 6b-d), albeit to a lesser extent, which may be explained by their decreased usage under suppressed CD8+ T-cell proliferation and associated DNA synthesis (Fig. 1b-c) or increased salvage.”

12- How was the 40mM concentration of aspartate decided? Again, upregulation of CD71 could be a differentiation marker? In line 410 authors claim that aspartate can overcome P53 cell cycle inhibition despite elevated CDKN1A. This is a strong statement but is lacking functional data to back it up. The same is the case with line 420, when it is mentioned that aspartate does not attenuate mitochondrial dysfunction(CI, CII). This could be shown by blocking the complexes or running a Seahorse assay.

We conducted multiple nutrient screens and 40 mM aspartate gave a robust effect – most publications use 20 mM aspartate such as this one Distinct modes of mitochondrial metabolism uncouple T cell differentiation and function | Nature which we observed also rescues iron deficient cells (see below, data not included in the manuscript), albeit not to the same extent as 40 mM aspartate. Since 40 mM gave a strong and consistent effect, we opted for this dose.

As described in response to a previous point, CD71 acts both as an activation marker and an iron response marker. However, in the context of low iron, the relative increase in CD71 is due to the decrease of iron availability.

We have toned down the sentence regarding *Cdkn1a*:

“Aspartate supplementation only marginally reduced Cdkn1a expression in low iron conditions (Supplementary Fig. 8d) indicating that the proliferative advantage conferred by aspartate is not entirely due to Cdkn1a suppression.”

We have removed the statement stating that aspartate is unable to attenuate mitochondrial dysfunction at CI and CIII based on difference in mROS production with or without aspartate supplementation as we agree that this is speculative.

Standard Seahorse assays do not inform about individual mitochondrial complex function and instead specialist setups are required. However, the defects we observe are in the TCA cycle which lies upstream of the ETC and consequently, a specialised Seahorse assay seems unwarranted. As we mentioned above, previous work from our group (Hepcidin-Mediated Hypoferremia Disrupts Immune Responses to Vaccination and Infection: Med) which utilised the same iron deficiency model in CD8⁺ T-cells showed that mitochondrial ATP production measured via the Seahorse ATP rate kit was reduced in low iron.

Comments on figures:

1- Figure 3b: the colour scheme is very confusing and the figure is hard to follow. It is not mentioned how the samples were normalized or whether we are looking at the gray or blue portions.

We have removed the top half of the figure (lactate, serine, alanine, glycine, pyruvate) as we don't believe this was relevant for the points we were making. We have also removed some of the details of the figure to improve clarity. The data shown in this plot is the metabolite abundance normalised by using a spiked in glutaric acid standard. We show the labelled fraction in blue and the unlabelled fraction in grey. These are stacked bar graphs where the sum (height of the stacked bars) denotes the total amount of metabolite detected. The most important aspects of this figure are (a) the labelled (blue) fraction which shows how much glutamine is being utilised in the TCA cycle and (b) the total height of the bars which shows how much metabolite is present. We have updated the figure legend with these details accordingly. We have also used open circles to denote unlabelled carbon atoms.

2- Figure 5g: the labeling does not match the text for the oxidative/reductive cycles.

Thank you, this has now been corrected.

3- Figure 6g is missing the colour legends.

The legend above Fig. 6c applies to all plots below.

4- Line 393 onwards: I believe you mean Fig 6 and not 7.

Reviewer #4 (Remarks to the Author):

- 1 Frost, J. N. *et al.* Hepcidin-Mediated Hypoferremia Disrupts Immune Responses to Vaccination and Infection. *Med (N Y)* **2**, 164-179.e112, doi:10.1016/j.medj.2020.10.004 (2021).
- 2 Wilkinson, N. & Pantopoulos, K. The IRP/IRE system in vivo: insights from mouse models. *Front Pharmacol* **5**, 176, doi:10.3389/fphar.2014.00176 (2014).
- 3 Sinclair, L. V. *et al.* Control of amino-acid transport by antigen receptors coordinates the metabolic reprogramming essential for T cell differentiation. *Nat Immunol* **14**, 500-508, doi:10.1038/ni.2556 (2013).
- 4 Yarosz, E. L. *et al.* Cutting Edge: Activation-Induced Iron Flux Controls CD4 T Cell Proliferation by Promoting Proper IL-2R Signaling and Mitochondrial Function. *J Immunol* **204**, 1708-1713, doi:10.4049/jimmunol.1901399 (2020).
- 5 Berg, V. *et al.* Iron Deprivation in Human T Cells Induces Nonproliferating Accessory Helper Cells. *Immunohorizons* **4**, 165-177, doi:10.4049/immunohorizons.2000003 (2020).
- 6 Fu, D. & Richardson, D. R. Iron chelation and regulation of the cell cycle: 2 mechanisms of posttranscriptional regulation of the universal cyclin-dependent kinase inhibitor p21CIP1/WAF1 by iron depletion. *Blood* **110**, 752-761, doi:10.1182/blood-2007-03-076737 (2007).
- 7 Shapiro, J. S. *et al.* Iron drives anabolic metabolism through active histone demethylation and mTORC1. *Nat Cell Biol* **25**, 1478-1494, doi:10.1038/s41556-023-01225-6 (2023).
- 8 An, W. G. *et al.* Stabilization of wild-type p53 by hypoxia-inducible factor 1alpha. *Nature* **392**, 405-408, doi:10.1038/32925 (1998).
- 9 Henning, A. N., Roychoudhuri, R. & Restifo, N. P. Epigenetic control of CD8(+) T cell differentiation. *Nat Rev Immunol* **18**, 340-356, doi:10.1038/nri.2017.146 (2018).
- 10 McNeill, L. A. *et al.* Hypoxia-inducible factor prolyl hydroxylase 2 has a high affinity for ferrous iron and 2-oxoglutarate. *Mol Biosyst* **1**, 321-324, doi:10.1039/b511249b (2005).
- 11 Andreini, C., Putignano, V., Rosato, A. & Banci, L. The human iron-proteome. *Metallomics* **10**, 1223-1231, doi:10.1039/c8mt00146d (2018).
- 12 Bayeva, M. *et al.* mTOR regulates cellular iron homeostasis through tristetraprolin. *Cell Metab* **16**, 645-657, doi:10.1016/j.cmet.2012.10.001 (2012).
- 13 Firkin, F. a. R., B. Interpretation of biochemical tests for iron deficiency: diagnostic difficulties related to limitations of individual tests. *Aust Prescr* **20**, 74-76, doi:<https://doi.org/10.18773/austprescr.1997.063> (1997).
- 14 Ubellacker, J. M. *et al.* Lymph protects metastasizing melanoma cells from ferroptosis. *Nature* **585**, 113-118, doi:10.1038/s41586-020-2623-z (2020).
- 15 Vousden, K. H. & Ryan, K. M. p53 and metabolism. *Nat Rev Cancer* **9**, 691-700, doi:10.1038/nrc2715 (2009).
- 16 Wang, Z. *et al.* Iron Drives T Helper Cell Pathogenicity by Promoting RNA-Binding Protein PCBP1-Mediated Proinflammatory Cytokine Production. *Immunity* **49**, 80-92.e87, doi:10.1016/j.immuni.2018.05.008 (2018).
- 17 Shyer, J. A., Flavell, R. A. & Bailis, W. Metabolic signaling in T cells. *Cell Res* **30**, 649-659, doi:10.1038/s41422-020-0379-5 (2020).
- 18 Howden, A. J. M. *et al.* Quantitative analysis of T cell proteomes and environmental sensors during T cell differentiation. *Nat Immunol* **20**, 1542-1554, doi:10.1038/s41590-019-0495-x (2019).
- 19 Kaelin, W. G., Jr. ROS: really involved in oxygen sensing. *Cell Metab* **1**, 357-358, doi:10.1016/j.cmet.2005.05.006 (2005).

REVIEWER COMMENTS

Reviewer #1 (Remarks to the Author):

I appreciate the author's efforts to improve the manuscript, but I have one comment. For Figure 4F, the author should compare the H3K27Me3 levels on specific gene loci (such as CD25, CD71, CD98, IL-2, TNF, and IFN γ) under low versus high iron conditions.

Compared to H3K4me3 and H3K27ac which are typically well enriched around promoters, the generally repressive mark H3K27me3 is often more dispersed across gene loci and present at lower levels. Since H3K27me3 is not as enriched at specific genomic features the background signal can be quite high. Further, we conducted our H3K27me3 ChIPmentation experiment on relatively few cells leading to low signal overall, and often particularly low on actively transcribed loci, such as the T-cell activation genes mentioned by the reviewer. Together, these factors make it hard to visually observe subtle changes in tracks at lowly H3K27 trimethylated regions. Even so, if we look at the region surrounding the *Tfrc* locus (encoding TFR1/CD71) we can still observe an enrichment of H3K27me3 in the low iron condition. In regions not activated in T-cells and hypermethylated with H3K27me3 such as the *Hoxa* and *Hoxd* regions, elevated levels of H3K27me3 are more easily observable in the low iron condition (see below, data not included in the manuscript), consistent with a general effect of low iron increasing H3K27me3.

This current study aims to establish iron deficiency's impacts on cellular state at a metabolic and epigenetic level. We believe the quantification analysis in Fig. 4e-f show low iron conditions do cause a global change in H3K27me3 abundance at the chromatin level, and the tracks below are consistent with this. However, we do not claim any gene-specific effects for H3K27me3 in the manuscript. A more in-depth exploration of low iron's impacts on T-cell chromatin including H3K27me3 at a gene locus resolution, and relating that to activation trajectory and differentiation, would be an interesting question, but is not the subject of this manuscript.

Reviewer #2 (Remarks to the Author):

The revised manuscript is substantially improved and has addressed key issues raised by the reviewers.

Reviewer #3 (Remarks to the Author):

The diligent point by point answers and additional data provided in this rebuttal are appreciated, and the inclusion of the comprehensive activation/function panel related to T cell function strengthened the author's claims that iron deficiency dampens T cell function.

Unfortunately this reviewer still thinks that the often strong mechanistic claims made by the authors are not fully backed up by the data presented, both the original and additional data, resulting in limited enthusiasm for publication of this study as is in a high impact broad readership journal.

The interpretation of the metabolic data remains incomplete and in places is used to stretch the conclusions based on the limited assays performed.

The authors claim that there is lower oxidative TCA cycle activity and increased reductive TCA cycle, to circumvent iron dependent SDH and ACO2. The data provided for this falls short to support this point. The authors report “M+3 succinate from the reductive TCA cycle”. The reductive TCA cycle flux is often reported as m+4 citrate from U13C GLN labelling, or m+3 malate and/or aspartate from U13C glucose labelling. M+3 succinate can be generated from the oxidative flux of the TCA as well, and as suggested, time course experiments would be helpful to unravel this. Long term labelling to achieve stable incorporation can cause scrambling of carbons in many intermediary metabolites making this interpretation difficult, especially when looking at directional incorporation such as directional fluxes in the TCA cycle (the removal of ‘flux’ to describe this data is appreciated).

We would first like to highlight that the statement quoted above (“M+3 succinate from the reductive TCA cycle”) appears in the rebuttal rather than the manuscript itself. In that statement, we were clarifying that M+3 succinate could potentially be derived from reductive cycling of ¹³C6-glucose derived carbons. We acknowledge that M+3 succinate may also occur due to alternative labelling routes. However, within the manuscript, M+3 succinate labelling from ¹³C6-glucose is not used as evidence of a partial reversal of the TCA cycle to the reductive trajectory. Rather, as the reviewer suggests, we predominantly utilise ¹³C6-glucose labelling into M+3 fumarate, malate and aspartate and show a clear increase in these metabolites in the low iron conditions (Fig. 5h). M+3 succinate from ¹³C6-glucose is only used to calculate pyruvate carboxylase activity [(malate M+3 – succinate M+3)/pyruvate M+3] which is a metric derived from this publication: Tumor cells dictate anti-tumor immune responses by altering pyruvate utilization and succinate signaling in CD8+ T cells: Cell Metabolism.

Further, we acknowledge that a full reversal of the TCA cycle to the reductive trajectory is unlikely and do not wish to claim this. Rather, the data indicate there is increased metabolism of glucose via pyruvate carboxylase into the metabolites fumarate, malate and aspartate, as measured via ¹³C6-glucose labelling into M+3 isotopomers (Fig. 5h). We have revised the manuscript to stress that the data only provides evidence of a partial reversal to the reductive trajectory between oxaloacetate to fumarate and we have modified the schematic in Fig. 5g to reflect this. We have revised the following sentences:

“This aspartate accumulation could be somewhat explained by increased production by an alternative pathway, for instance pyruvate anaplerosis to oxaloacetate via pyruvate carboxylase (PC) and a partial reversal of TCA cycling (Fig. 5g)⁴⁹.”

“During iron-deficiency, significant increases in ¹³C6-glucose labelling into M+3 TCA cycle metabolites including fumarate and malate were observed, indicative of increased PC

contribution to these metabolites and reductive TCA cycling from oxaloacetate to fumarate (Fig. 5h).”

“A partial reversal of TCA cycle activity is also in agreement with the observed increase in the NAD⁺/NADH ratio (Fig. 3g, Supplementary Fig. 4h). This suggests that under iron limiting conditions, CD8⁺ T-cells may potentially utilise PC and a partial reversal of the TCA cycle to replenish the depleted metabolites fumarate and malate, circumventing the use of the iron-dependent enzymes, ACO2 and SDH.”

Fig. 5g:

We agree with the reviewer that “time course experiments would be helpful to unravel [the succinate labeling pattern]”. However, we would also highlight that CD8⁺ T cells express low levels of the key glucose transporters, Glut1 (SLC2A1) and Glut3 (SLC2A3) at early time points following stimulation, which begin to upregulate at ~20 hours of activation (see below, data not included in the manuscript). It would therefore be challenging to quantify distinct mass isotopomers of glucose metabolites at early time points, when overall ¹³C6-glucose uptake is likely to be very low. In the data presented, after 24 hours of T cell activation in presence of ¹³C6-glucose, overall fractional labeling of succinate and other TCA cycle metabolites is just 20%.

The fact that cell counts can be a huge challenge to overcome when planning experiments with primary cells is not lost on this reviewer, but if the authors are not able to provide the data to support their conclusions, it becomes an assumption, leading to potential misinterpretations.

The authors did alter the statement that “aspartate supplementation confers a proliferative advantage was altered”, to tone down the claimed that aspartate can overcome P53 cell cycle inhibition despite elevated CDKN1A. Similarly, authors removed the statement that aspartate does not attenuate mitochondrial dysfunction at CI and CII for being speculative, however, a seahorse assay to validate this claim could have supported this with more mechanistic insight, which the authors preferred to leave out.

As the authors highlight in the rebuttal, aspartate supplementation and the subsequent phenotype of higher glycolytic rate and ATP production are not iron dependent observations. Conversely, the manuscript still claims (even in the title) that iron deficiency causes aspartate-sensitive metabolic and proliferative dysfunction in T cells.

Thus, although there are interesting changes that are associated with altered mitochondrial function in iron deficient T cells, the concern about mechanistic claims that are not fully backed up by the data remain.

We acknowledge that aspartate has some effects on CD8+ T-cell metabolism and function that are independent of iron availability. However, we also maintain that aspartate modifies certain important CD8+ T-cell activities in an iron dependent manner. For instance, aspartate boosts proliferation in iron starved cells greater than in iron replete cells (Fig. 6b-d). Aspartate also modifies TFR1/CD71 expression and H3K27me3 abundance in an iron-dependent manner (Fig. 6e and 6l) with differential aspartate effects depending on whether iron levels are high or low. However, the reviewer is correct that some aspects of metabolism are improved by aspartate in both iron low and iron replete cells. To avoid suggesting that aspartate alters metabolic remodelling in general only under iron deficiency, we have modified and simplified the title to *"Iron deficiency causes aspartate-sensitive inhibition of CD8+ T cells"*.

We also modified the following sentence:

“Remarkably, aspartate supplementation of culture media, assumed to access the cytosol directly, profoundly rescued iron-deficiency impaired proliferation and enhanced other functional aspects of activated T-cells in both iron dependent and independent ways.”

Reviewer #4 (Remarks to the Author):

REVIEWERS' COMMENTS

Reviewer #1 (Remarks to the Author):

I am satisfied with the author's effort to improve the manuscript.

Reviewer #3 (Remarks to the Author):

I appreciate the detailed answers, and the text and title modification to better describe the data being shown.

I understand that revision rounds are exhausting when the story might already seem so clear to the authors, but it would have further strengthened the study if some additional supporting experiments would have been performed to further cement the metabolic findings.

This reviewer still struggles with the interpretation of the "reductive TCA" and especially the m+3 citrate label, since that could well be coming from oxidative TCA through PC activity, especially given the larger relative increase in the m+5 pools compared to the m+3 pools in panel 5H. At the very least the authors should show the labeling in the full range of isotopomers (m+0 through m+6) of citrate in that context.

As the reviewer has requested, we have included the full isotopomers dataset for the ¹³C6-glucose tracing for citrate, along with the following text:

"During iron-deficiency, significant increases in ¹³C6-glucose labelling into M+3 TCA cycle metabolites including fumarate and malate were observed, indicative of increased PC contribution to these metabolites and reductive TCA cycling from oxaloacetate to fumarate (Fig. 5h). M+3 and M+5 citrate isotopomers are also likely derived from PC routed ¹³C6-glucose atoms (Fig. 5h, Supplementary Fig. 6e)"

Supplementary Fig. 6e:

Reviewer #4 (Remarks to the Author):
